# Keratoconus-susceptibility gene identification by corneal thickness genome-wide association study and artificial intelligence IBM Watson

Yoshikatsu Hosoda[1,2], Masahiro Miyake [1,2 ✉], Akira Meguro[3], Yasuharu Tabara[2], Sachiko Iwai[1], Naoko Ueda-Arakawa[1], Eri Nakano[1,2], Yuki Mori[1,2], Munemitsu Yoshikawa[1,2], Hideo Nakanishi[1], Chiea-Chuen Khor [4], Seang-Mei Saw[4,5,6], Ryo Yamada[2], Fumihiko Matsuda[2], Ching-Yu Cheng [4,6,7], Nobuhisa Mizuki[3], Akitaka Tsujikawa[1], Kenji Yamashiro [1,8] & The Nagahama Study Group*

Keratoconus is a common ocular disorder that causes progressive corneal thinning and is the leading indication for corneal transplantation. Central corneal thickness (CCT) is a highly heritable characteristic that is associated with keratoconus. In this two-stage genome-wide association study (GWAS) of CCT, we identified a locus for CCT, namely *STON2* rs2371597 ($P = 2.32 \times 10^{-13}$), and confirmed a significant association between *STON2* rs2371597 and keratoconus development ($P = 0.041$). Additionally, strong *STON2* expression was observed in mouse corneal epithelial basal cells. We also identified *SMAD3* rs12913547 as a susceptibility locus for keratoconus development using predictive analysis with IBM's Watson question answering computer system ($P = 0.001$). Further GWAS analyses combined with Watson could effectively reveal detailed pathways underlying keratoconus development.

[1] Department of Ophthalmology and Visual Sciences, Kyoto University Graduate School of Medicine, Kyoto, Japan. [2] Center for Genomic Medicine, Kyoto University Graduate School of Medicine, Kyoto, Japan. [3] Department of Ophthalmology and Visual Science, Yokohama City University Graduate School of Medicine, Yokohama, Japan. [4] Singapore Eye Research Institute, Singapore National Eye Centre, Singapore, Singapore. [5] Saw Swee Hock School of Public Health, National University of Singapore, Singapore, Singapore. [6] Ophthalmology and Visual Sciences Academic Clinical Program, Duke-NUS Medical School, Singapore, Singapore. [7] Department of Ophthalmology, Yong Loo Lin School of Medicine, National University of Singapore, Singapore, Singapore. [8] Department of Ophthalmology, Otsu Red-Cross Hospital, Otsu, Japan. *A list of authors and their affiliations appears at the end of the paper. ✉email: miyakem@kuhp.kyoto-u.ac.jp

Keratoconus is a common, bilateral, noninflammatory type of corneal degeneration, affecting one out of every 375 people in the general population[1], and is a major indication for corneal transplantation in developed countries[2–4]. Keratoconus is a progressive eye disease characterised by an asymmetrical conical protrusion due to focal thinning of the cornea, which causes variable and severe visual disturbances. Thus, patients with severe keratoconus usually present with thin corneas, although not all patients with thin corneas develop the disease.

Although previous genome-wide association studies (GWASs) were performed with keratoconus patients, no genetic region with a significant genome-wide association has been identified thus far[5–7]. Most of keratoconus-susceptibility genes were identified through a large meta-GWAS of central corneal thickness (CCT), because CCT is closely associated with keratoconus and is highly heritable trait[8–11]. For example, Lu et al. conducted a GWAS on CCT ($n = 20,020$), screened for associations between 26 CCT-associated loci and keratoconus ($n_{case} = 874$, $n_{control} = 6085$), and identified six loci associated with keratoconus susceptible[12]. When the sample size of patients with disease is limited, this intermediate phenotype strategy is reasonable because it is less affected by multiple testing correction, when compared to a simple GWAS. However, where possible, it is preferable to reasonably narrow down the number of candidate genes that are screened, in order to avoid 'the curse of multiplicity'. Indeed, although a recent large meta-GWAS of CCT ($n = 25,910$) identified multiple additional CCT-associated loci, no novel keratoconus-susceptible genes were identified through sequential screening ($n_{case} = 933$, $n_{control} = 5946$)[13]. In addition, pathway analysis of keratoconus has not been performed because of the lack of reliable GWASs of keratoconus. To reveal the specific role of various genes in keratoconus development, it may be important to identify susceptibility genes with relatively smaller effect size, which cannot be identified through large sample size GWAS studies[14].

In this study, we used the latest artificial intelligence technology, IBM's Watson for Drug Discovery (WDD), to select candidate genes from the results of our newly performed GWAS on CCT. WDD, which has been configured to support life science research, can help in understanding and identifying relationships between molecules/genes, using various types of enormous datasets from structured databases and PubMed database[15,16]. To identify new keratoconus-susceptibility loci, we performed a traditional two-stage GWAS on CCT using a Japanese community-based cohort, and assessed the association of CCT-susceptibility loci with keratoconus occurrence. Further, we assessed new keratoconus-susceptibility genes derived from the combination of GWAS results and WDD.

## Results

**Two-stage GWAS of CCT in Japanese patients.** To investigate and identify genetic loci associated with CCT, the average CCT in both eyes was used as the dependent variable for genome-wide quantitative trait locus (QTL) analyses. We included the age, sex, average axial length of both eyes, and the first principal component as covariates. An inflation factor ($\lambda_{GC}$) of 1.065 indicated good control of the study population substructure (Supplementary Fig. 1). We also found that inclusion of additional PCs minimally altered the findings.

During the first stage with 3584 participants, we identified two single-nucleotide polymorphisms (SNPs), namely *STON2* rs2371597 ($\beta = 5.35$, $P = 1.91 \times 10^{-11}$) and *FNDC3B* rs4894538 ($\beta = -4.13$, $P = 1.90 \times 10^{-8}$), which exceeded genome-wide significance (Fig. 1 and Table 1). Since *FNDC3B* rs4894538 was in strong linkage disequilibrium with rs4894535 ($R^2 = 0.991$ in the discovery set), which is an established SNP associated with CCT and keratoconus, we only carried *STON2* rs2371597 forward to the replication stage. In the replication stage, we analysed 2942 participants and found that *STON2* rs2371597 showed a significant association with CCT ($\beta = 2.95$, $P = 3.58 \times 10^{-4}$). Meta-analysis of the discovery and replication sets further confirmed a robust association of *STON2* rs2371597 with CCT ($\beta_{meta-Japanese} = 4.20$, $P_{meta-Japanese} = 2.32 \times 10^{-13}$).

The baseline characteristics of the participants and SNPs that showed $P$-values of $<1.0 \times 10^{-4}$ are shown in the Supplementary Table 1 and Supplementary Data 1.

**Replication with patients of other ethnicities.** To confirm the association between *STON2* rs2371597 and CCT in individuals with other ethnicities, we conducted trans-ethnic replication analyses using three different cohorts comprised of Malay, Chinese, and Indian subjects ($N = 2510$, 2469, and 2508, respectively) (Table 2). In all cohorts, the rs2371597 C allele was associated with increased CCTs ($\beta = 2.14$, $P = 0.0406$, Malay cohort; $\beta = 2.44$, $P = 0.021$, Chinese cohort; $\beta = 1.56$, $P = 0.130$, Indian cohort), which was in agreement with the data from the Japanese cohort. Thus, meta-analysis of the data from all ethnic groups revealed a strong association between rs2371597 and CCT ($P_{meta-all} = 2.18 \times 10^{-4}$). Additionally, publicly available GWAS data enabled us to discover an association of the *STON2* rs2371597 C allele with increased CCTs in a Latino population ($P = 6.12 \times 10^{-3}$, Table 2).

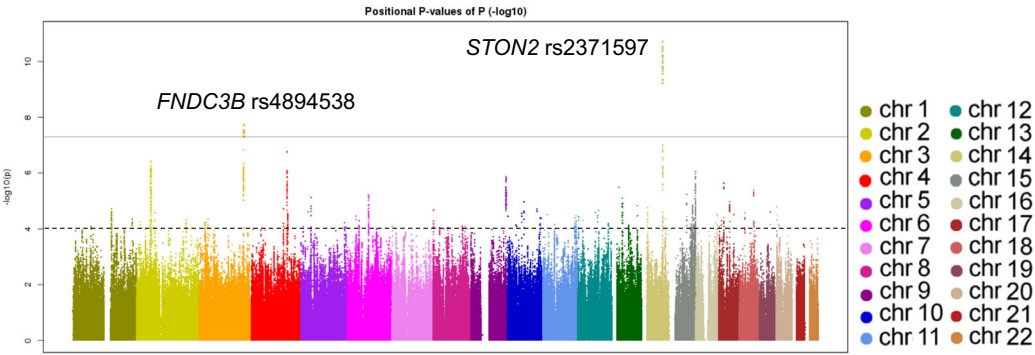

**Fig. 1 Manhattan plot of QTL data from our GWAS of CCT.** The plot shows $-\log_{10}$-transformed $P$-values for all SNPs adjusted for age, sex, average axial length in both eyes, and the first principal component. The horizontal line represents the genome-wide significance threshold of $5.0 \times 10^{-8}$, and the lower dashed line represents the threshold of $1.0 \times 10^{-4}$ to identify candidate genes for Watson Drug Discovery analysis.

**Table 1 SNPs that showed genome-wide significant association with CCT in the discovery stages.**

| SNP | Chr | Position | Effect allele | EAF | Gene | Discovery stage | | | Replication stage | | | Meta-analysis | | |
|---|---|---|---|---|---|---|---|---|---|---|---|---|---|---|
| | | | | | | N | β | P | N | β | P | N | β | P |
| rs4894538 | 3 | 171999005 | T | 0.299 | FNDC3B (intron) | 3584 | −4.13 | $1.90 \times 10^{-8}$ | – | – | – | – | – | – |
| rs2371597 | 14 | 81873377 | C | 0.242 | STON2 (intron) | 3584 | 5.35 | $1.91 \times 10^{-11}$ | 2942 | 2.95 | $3.58 \times 10^{-4}$ | 6526 | 4.20 | $2.32 \times 10^{-13}$ |

Chr, chromosome; EAF, effect allele frequency; β, beta; P, P-value.

**Table 2 Replication of the STON2 rs2371597 association with CCT.**

| SNP | Chr | Position | Effect allele | Cohort | Ethnicity | N | Discovery stage | | |
|---|---|---|---|---|---|---|---|---|---|
| | | | | | | | β | SE | P |
| rs2371597 | 14 | 81873377 | C | Nagahama | Japanese | 6526 | 4.20 | 0.573 | $2.32 \times 10^{-13}$ |
| | | | C | SiMES | Malay | 2510 | 2.14 | 1.045 | 0.0406 |
| | | | C | SCES | Chinese | 2469 | 2.44 | 1.063 | 0.0215 |
| | | | C | SINDI | Indian | 2508 | 1.56 | 1.028 | 0.130 |
| | | | C | 4 collections | Meta | 14,013 | 3.17 | 0.415 | $2.18 \times 10^{-14}$ |
| | | | C | – | Latino | 3584 | –[a] | – | $6.12 \times 10^{-3}$ |

SE, standard error.
[a]The effect direction was the same as those found with the other ethnicities.

**Association of previously reported CCT-associated loci with CCT.** We also evaluated the association of 93 previously reported CCT-susceptible SNPs from 57 genetic regions in our discovery GWAS, using 3584 samples[10,11]. Of these SNPs, 53 were included in our discovery dataset, and 30 (56.6%) from 18 genetic regions showed an association with CCT with the same effect directions reported previously ($P < 0.05$, Supplementary Data 2). After Bonferroni's correction, 14 of the 53 SNPs (26.4%) showed a significant association with CCT. Additionally, we assessed the associations of SNPs with the lowest P-value in the 48 genetic regions. We found 39 of the top 48 SNPs (81.3%) had a P-value of <0.05, and 16 SNPs (33.3%) showed significant associations with CCT after Bonferroni's correction (Supplementary Data 2).

**Association of STON2 with keratoconus.** To evaluate the possible association between STON2 rs2371597 and keratoconus, we conducted a case-controlled study using 179 keratoconus patients, regardless of age, sex, or physical impairment, from the Yokohama City University and 11,084 Japanese healthy controls. This analysis revealed that minor-allele frequency (MAF) at rs2371597 C was significantly high in keratoconus patients than that in controls ($MAF_{keratoconus} = 0.293$, $MAF_{controls} = 0.246$, odds ratio [OR] (95% confidence interval [CI]) = 1.27 (1.01–1.60), $P = 0.041$).

**Expression of STON2 in human tissues and in the mouse cornea.** A search of a publicly available expression quantitative trait loci analysis (eQTL) database revealed that rs2371597 was significantly associated with STON2 expression (GTEx Portal; https://gtexportal.org/home/). A multi-tissue eQTL plot revealed that the normalised effect size (NES) of rs2371597 on STON2 expression was strongest in the skeletal muscle (NES = −0.189, $P = 1.3 \times 10^{-8}$; Fig. 2, https://www.gtexportal.org/home/snp/rs2371597) in which collagen plays an important role in providing its tensile strength and elasticity. However, data on the association of rs2371597 with gene expression in human corneal tissue were not available.

Our immunohistochemical study of mouse corneas showed that STON2 was expressed in the corneal epithelial cell layer (Fig. 3). Our results demonstrated that strong STON2 expression mainly occurred in basal cells rather than superficial cells in the corneal epithelium. In the stroma and endothelium layer, only minimal STON2 expression was observed.

**Predictive analysis using the WDD system.** WDD predictive analysis scores and visualisation of the relationships between 'teaching genes' and candidate genes (based on searching millions of Medline abstracts in PubMed [accessed on March 1st, 2018], as described in the Methods section). The similarity score (i.e. the degree of similarity to 'teaching genes') of each candidate gene is shown in Supplementary Table 2. The tree plot and scatter plot results are shown in Figs. 4 and 5. Three major streams were observed, and STON2 was part of the FOXO1–SMAD3 stream (Fig. 4).

We focused on candidate genes within the FOXO1–SMAD–STON2 stream. Specifically, we selected seven keratoconus-susceptibility genes (CDH13, SMAD3, ADAM12, CSMD1, NRXN1, CPLX2, and WWOX) for further analysis. Among these candidate genes, SMAD3 showed the highest similarity score when compared to the 'teaching genes' (Supplementary Table 3). Since the SMAD3 rs11333560 genotype was not available in the Tohoku Medical Megabank Project, we instead employed a proxy SNP (rs12913547 in SMAD3). The SMAD3 rs12913547 T allele was significantly associated with an increased risk for keratoconus development after Bonferroni's correction (OR (95% CI) = 1.44 (1.16–1.80), $P = 0.001$, Table 3). CPLX2 rs4242187 tended to show an association with keratoconus, but the association was not statistically significant (OR (95% CI) = 0.70 (0.46–1.05), $P = 0.082$).

## Discussion

In this study, we identified two keratoconus-susceptibility genes by integrating conventional GWAS of CCT and artificial intelligence, using the WDD platform. We found that the STON2 rs2371597 allele was significantly associated with CCT and keratoconus, as well as significantly altered STON2 expression in

| Tissue | Samples | NES | p-value | m-value |
|---|---|---|---|---|
| Minor Salivary Gland | 85 | 0.167 | 0.08 | 0.00600 |
| Colon - Transverse | 246 | 0.109 | 0.03 | 0.00 |
| Ovary | 122 | 0.0999 | 0.4 | 0.0680 |
| Pituitary | 157 | 0.0918 | 0.4 | 0.0620 |
| Prostate | 132 | 0.0733 | 0.4 | 0.0410 |
| Brain - Amygdala | 88 | 0.0652 | 0.4 | 0.0160 |
| Thyroid | 399 | 0.0297 | 0.5 | 0.00 |
| Brain - Frontal Cortex (BA9) | 118 | 0.0256 | 0.6 | 0.0100 |
| Skin - Not Sun Exposed (Suprapubic) | 335 | 0.0244 | 0.6 | 0.00100 |
| Uterus | 101 | 0.0204 | 0.9 | 0.170 |
| Brain - Hippocampus | 111 | 0.00855 | 0.9 | 0.0360 |
| Heart - Left Ventricle | 272 | 0.00693 | 0.9 | 0.00100 |
| Small Intestine - Terminal Ileum | 122 | 0.00664 | 0.9 | 0.149 |
| Whole Blood | 369 | 0.00657 | 0.9 | 0.00 |
| Adipose - Visceral (Omentum) | 313 | 0.00569 | 0.9 | 0.0130 |
| Brain - Substantia nigra | 80 | 0.00432 | 1 | 0.0870 |
| Brain - Putamen (basal ganglia) | 111 | 0.00168 | 1 | 0.0280 |
| Cells - EBV-transformed lymphocytes | 117 | - | - | - |
| Brain - Nucleus accumbens (basal ganglia) | 130 | -0.000478 | 1 | 0.0290 |
| Stomach | 237 | -0.00426 | 0.9 | 0.0360 |
| Brain - Anterior cingulate cortex (BA24) | 109 | -0.00519 | 0.9 | 0.0380 |
| Spleen | 146 | -0.00873 | 0.9 | 0.0630 |
| Breast - Mammary Tissue | 251 | -0.0132 | 0.8 | 0.0860 |
| Brain - Caudate (basal ganglia) | 144 | -0.0211 | 0.7 | 0.0180 |
| Cells - Transformed fibroblasts | 300 | -0.0233 | 0.6 | 0.0190 |
| Lung | 383 | -0.0239 | 0.6 | 0.0670 |
| Heart - Atrial Appendage | 264 | -0.0300 | 0.5 | 0.0240 |
| Vagina | 106 | -0.0313 | 0.6 | 0.0690 |
| Artery - Coronary | 152 | -0.0345 | 0.5 | 0.102 |
| Liver | 153 | -0.0392 | 0.4 | 0.169 |
| Brain - Cortex | 136 | -0.0452 | 0.4 | 0.161 |
| Testis | 225 | -0.0488 | 0.4 | 0.150 |
| Skin - Sun Exposed (Lower leg) | 414 | -0.0638 | 0.1 | 0.197 |
| Esophagus - Gastroesophageal Junction | 213 | -0.0706 | 0.2 | 0.271 |
| Nerve - Tibial | 361 | -0.0803 | 0.04 | 0.360 |
| Esophagus - Mucosa | 358 | -0.103 | 0.3 | 0.838 |
| Pancreas | 220 | -0.107 | 0.3 | 0.268 |
| Brain - Hypothalamus | 108 | -0.109 | 0.1 | 0.444 |
| Esophagus - Muscularis | 335 | -0.110 | 0.006 | 0.847 |
| Artery - Aorta | 267 | -0.119 | 0.03 | 0.741 |
| Colon - Sigmoid | 203 | -0.153 | 0.03 | 0.841 |
| Brain - Cerebellum | 154 | -0.162 | 0.04 | 0.700 |
| Brain - Cerebellar Hemisphere | 125 | -0.162 | 0.04 | 0.774 |
| Brain - Spinal cord (cervical c-1) | 83 | -0.177 | 0.2 | 0.412 |
| Muscle - Skeletal | 491 | -0.181 | 1.1e-5 | 1.00 |
| Artery - Tibial | 388 | -0.185 | 2.6e-5 | 0.994 |
| Adipose - Subcutaneous | 385 | -0.188 | 4.6e-6 | 1.00 |
| Adrenal Gland | 175 | -0.238 | 0.01 | 0.760 |

**Single-tissue eQTL NES (with 95% CI)** — x-axis: NES (-0.4, -0.2, 0.0, 0.2)

**Single-tissue eQTL p-value versus Multi-tissue Posterior Probability** — y-axis: -log10( Single-tissue eQTL p-value ); x-axis: m-value (Posterior Probability from METASOFT) (0.0, 0.2, 0.4, 0.6, 0.8, 1.0)

**Fig. 2 Multi-tissue eQTL-based comparison of the association between rs2371597 and *STON2* expression.** Multi-tissue eQTL plot generated by performing an eQTL search of a publicly available database (GTEx Portal; https://gtexportal.org/home/). The effect size of rs2371597 on *STON2* expression was strongest in the skeletal muscle (NES = −0.189, $P = 1.3 \times 10^{-8}$), followed by the adrenal gland (NES = −0.175, $P = 0.03$).

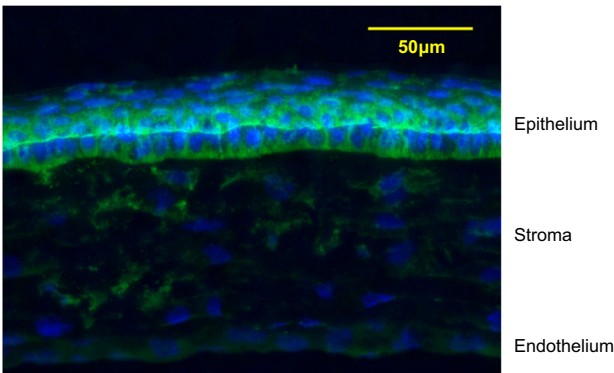

**Fig. 3 Expression of *STON2* in retinal ganglion cells of C57BL/6 mice.** Mouse corneal sections were immunostained with an antibody against *STON2* (green). The nuclei are counterstained with 40,6-diamidino-2-phenylindole (blue) in the merged image. The upper portion of the panel shows the corneal epithelium.

human tissues. WDD analysis assisted in identifying another keratoconus-susceptibility SNP, i.e. *SMAD3* rs12913547. Considering that recent association analyses of keratoconus failed to identify novel keratoconus-susceptibility genes despite their large sample sizes[5,6,13], the current approach of combining GWAS and artificial intelligence data appears to have been quite efficient.

We first conducted a conventional GWAS of CCT with trans-ethnic replication. This analysis enabled us (1) to confirm *STON2* as a susceptibility gene for CCT and (2) to show that our Japanese population shared most of underlying genetic effects not only with other Asian populations, but also with European and Latino populations[12,17]. *ZNF469* rs9938149, the most established CCT-susceptibility SNP, was replicated in our GWAS (n = 3584, β = −3.35, $P = 7.45 \times 10^{-3}$). The relatively high *P*-value was due to its small MAF in Japanese subjects, compared to European ethnic groups ($\text{MAF}_{\text{Japanese}} = 0.034$, $\text{MAF}_{\text{European}} = 0.349$; data from the 1000 Genomes database).

We also identified *STON2* rs2371597 (located at chromosome14q31) as a keratoconus-susceptibility SNP. *STON2* was previously identified as an endocytic adaptor dedicated to the retrieval of surface-stranded synaptic vesicle proteins[18]. A previous genetic study reported that *STON2* SNPs were associated with CCT, but not with keratoconus development in Caucasians. Therefore, as the prevalence of keratoconus is known to be higher in the Asian population compared to Caucasians, we speculate that the genetic and pathological background of keratoconus differs between ethnicities. Moreover, as it is possible that certain keratoconus-susceptibility genes identified in Asians are not associated with keratoconus development in Caucasians, further studies are required to compare the effect size of keratoconus-susceptibility genes on keratoconus development across various ethnicities, which will serve to reveal ethnical differences.

The eQTL data revealed that the effect size of rs2371597 on *STON2* expression was strongest in skeletal muscle, which is rich

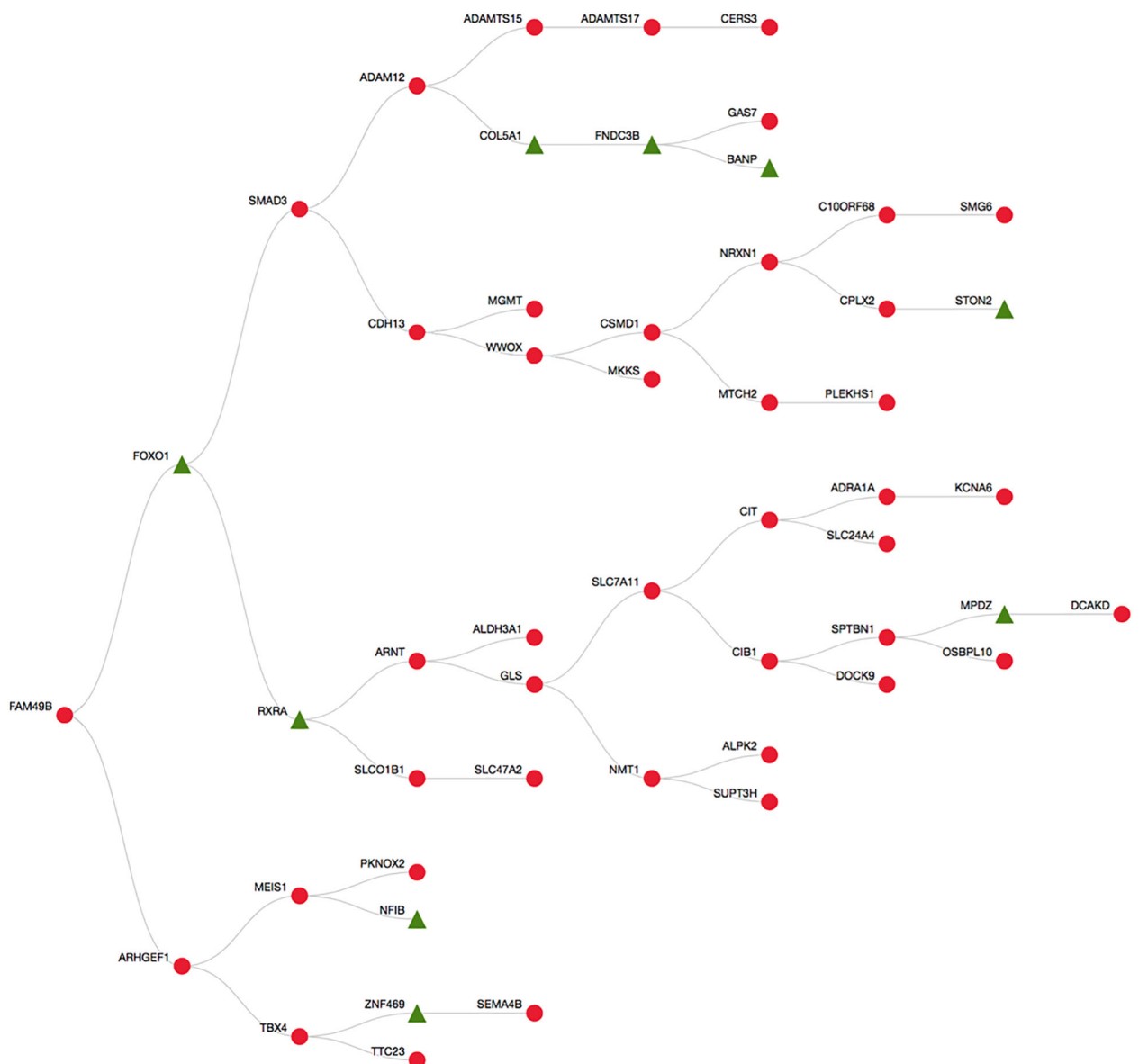

**Fig. 4 Tree plot of WDD-based predictive analysis.** Nine teacher genes (green triangles) and 42 candidate genes (red circles) are shown, based on predictive analysis with WDD. The relationships between the genes are visualised.

in collagen, a protein that reportedly plays a key role in keratoconus pathogenesis. Although no previous reports have investigated the expression of STON2 in human corneal tissue, we speculate that *STON2* may play an important role in keratoconus development by interacting with extracellular matrix remodelling. Our immunohistochemical staining results for STON2 are compatible with a previous histopathological study of keratoconus, which showed that thinning of the corneal stroma, breaks in Bowman's layer, and degeneration of the corneal epithelium were the characteristics of corneas in patients with keratoconus[2,19]. STON2 might be associated with the vulnerability to physical damage or immunological changes. Although biological studies examining the role of STON2 in human corneal tissue are required to prove our hypothesis, pathways involving *STON2* may serve as targets for treating keratoconus by controlling basal cell degeneration.

Through WDD, we identified *SMAD3* as an additional keratoconus-susceptibility gene (OR = 1.44 (1.16–1.80), *P* = 0.001), and *CPLX2* as a future candidate gene for keratoconus-

susceptibility (OR = 0.70 (0.46–1.05), *P* = 0.082). *SMAD3* is known to play a significant role in the transforming growth factor-β (TGF-β) signalling pathway, which may modulate extracellular matrix (ECM) alterations in keratoconus[20]. Data from a previous study showed TGF-β/SMAD3 signalling promoted gluconeogenesis through an interaction with *FOXO1*, which is an established keratoconus-susceptibility gene[21]. Further studies of the roles of *SMAD3*, *FOXO1*, and TGF-β signalling in keratoconus development are needed. In addition, *CPLX2* is another interesting target because (1) both *CPLX2* and *STON2* are reported to be associated with schizophrenia[18,22], and (2) comorbidity between schizophrenia and keratoconus was also demonstrated previously[23–26]. We speculate that schizophrenia and keratoconus employ shared underlying pathophysiological processes, which would be another pathway apart from gluconeogenesis. We expect that the relationships between these two diseases will be studied more in the future.

We expected that CCT-reducing alleles would be associated with an increased risk of developing keratoconus, given the

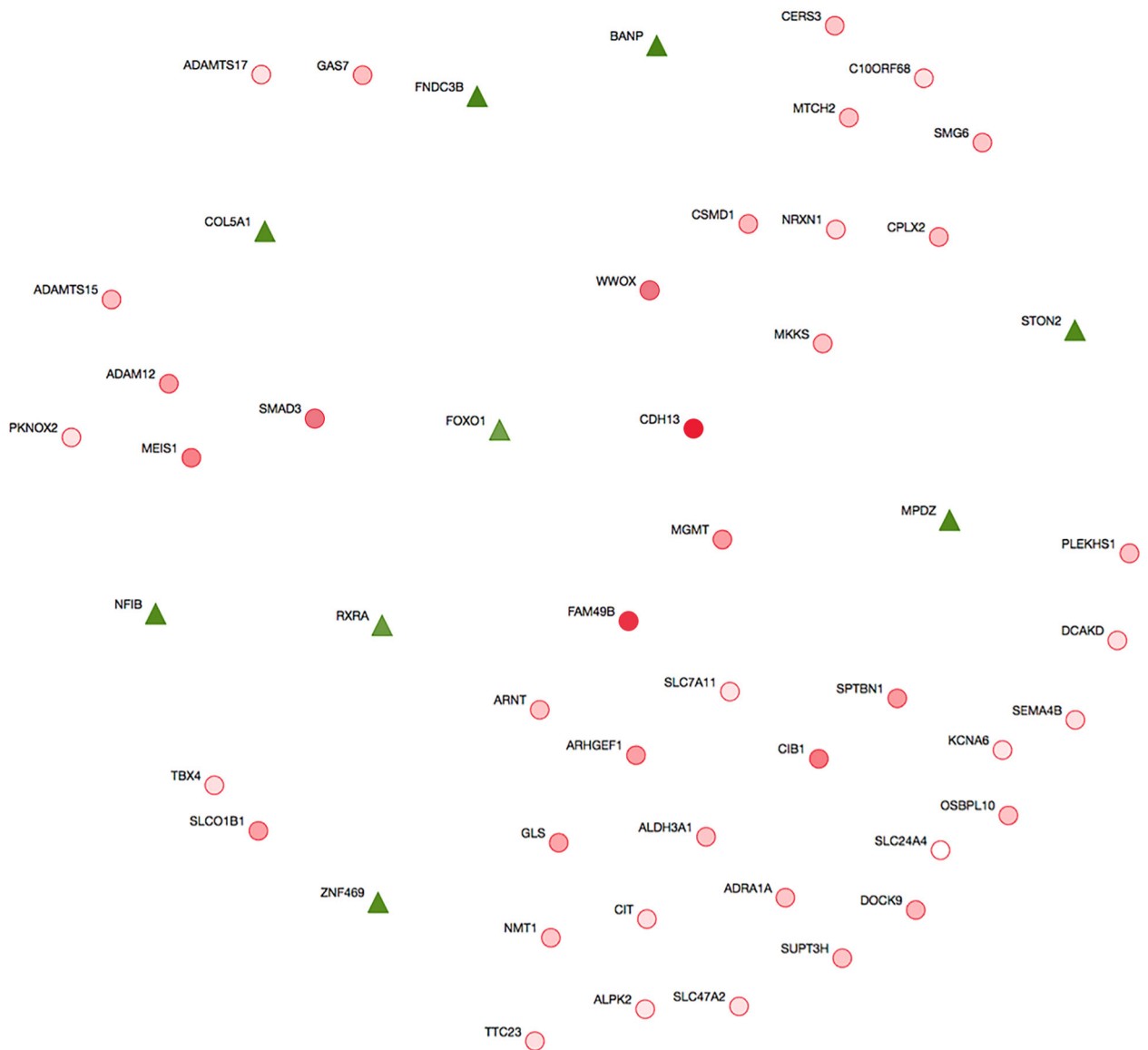

**Fig. 5 Scatter plot of WDD-based predictive analysis.** A scatter plot of genes was generated based on their distances in the literature. Nine teacher genes (green triangles) and 42 candidate genes (red circles) are shown.

**Table 3 Testing seven CCT-associated loci for keratoconus-susceptibility.**

| Gene | Chr | Position | SNP | Effect allele | Control | | Keratoconus | | Odds ratio (95% CI) | Pa |
|------|-----|----------|-----|---------------|---------|-----|-------------|-----|---------------------|-----|
| | | | | | N | EAF | N | EAF | | |
| NRXN1 | 2 | 50635086 | rs13382330 | C | 8137 | 0.176 | 179 | 0.159 | 0.89 (0.67–1.19) | 0.420 |
| CPLX2 | 5 | 175244097 | rs4242187 | T | 8150 | 0.097 | 179 | 0.070 | 0.70 (0.46–1.05) | 0.082 |
| CSMD1 | 8 | 2799288 | rs143428993 | A | 8148 | 0.012 | 179 | 0.017 | 1.39 (0.61–3.14) | 0.435 |
| ADAM12 | 10 | 127905962 | rs11244890 | A | 8156 | 0.211 | 179 | 0.196 | 0.91 (0.70–1.19) | 0.491 |
| SMAD3 | 15 | 67467507 | rs12913547 | T | 8096 | 0.563 | 179 | 0.651 | 1.44 (1.16–1.80) | 0.001b |
| WWOX | 16 | 78430702 | rs6564538 | T | 8156 | 0.110 | 179 | 0.117 | 1.07 (0.77–1.48) | 0.680 |
| CDH13 | 16 | 83650510 | rs1035533 | G | 8150 | 0.190 | 179 | 0.201 | 1.07 (0.83–1.39) | 0.594 |

Chr, chromosome; EAF, effect allele frequency; SNP, single-nucleotide polymorphism; CI, confidence interval.
[a]P-values derived using logistic-regression analysis.
[b]P-values are statistically significant after Bonferroni's correction.

known association between thin corneas and the disease. Indeed, most CCT-reducing alleles have been associated with an increased risk of keratoconus, and this was true for *SMAD3* rs12913547 in the current study. However, with *STON2*, a CCT-increasing allele led to increased keratoconus risk. Although this finding was unexpected, such complex association patterns have often been observed in previous studies. One example is that a body fat-decreasing allele near *IRS1* was also associated with

adverse lipid profiles, and increased risks for coronary artery disease and type-2 diabetes[27–29]. Another example is that the most famous CCT-reducing SNP, rs9938149, located between *BANP* and *ZNF469*, is known to decrease keratoconus risk[12]. We anticipate that further research on *ZNF469* and *STON2* will reveal pleiotropic effects for these genes in terms of corneal thickness and the risk for keratoconus development.

We applied a approach, i.e. WDD, to screen for keratoconus-susceptibility SNPs from thousands of potential CCT-susceptibility SNPs. While we focused on seven genes out of 52 candidate genes (based on the predictive analysis results of WDD), SNPs in only two genes showed a significant or marginal association with keratoconus. Since the similarity score was calculated based on Medline abstract searches in PubMed, genes that have been more extensively studied tend to have higher scores. We focused on genes located between 'teacher genes' and *STON2* in this study to identify genes that were not sufficiently evaluated in previous studies. Although *CPLX2* (ranked number 16 by Watson) showed a marginally significant association with keratoconus, only *SMAD3* (with the highest similarity score) showed a robust association with keratoconus. However, other genes with a relatively high similarity score, such as *CDH13* and *MEIS1*, were not associated with keratoconus. Interestingly, *CPLX2* and *SMAD3* were located next to teaching genes, whereas *CDH13* and *MEIS1* were not. We speculate that the rank of the similarity score may be an important index; however, the approach used to rank scores needs to be improved. To identify disease-susceptible genes with Watson, we propose that focusing on genes located between and/or next to teaching genes may be important. Further studies will be required to establish a method for combining GWAS with Watson.

In summary, we identified two keratoconus-susceptibility loci, *STON2* and *SMAD3*, by integrating conventional GWAS and artificial intelligence, using WDD. Cognitive-computing technology combined with GWAS can assist in identifying hidden relationships among disease-susceptibility genes and potential susceptibility genes, enabling more efficient interpretation of GWAS results. We believe that the current approach can be generalised for application to numerous other diseases. Since samples from patients with a disease are more difficult to obtain than samples from healthy individuals, which can be collected through cohort studies, the current approach will prove particularly helpful in facilitating the exploration of disease-susceptible genes.

## Methods

**Participants in the two-staged GWAS of CCT.** For the two-staged GWAS, we analysed data from healthy Japanese volunteers enrolled in the Nagahama Prospective Cohort for Comprehensive Human Bioscience (the Nagahama Study). The initial follow-up data were collected from 9850 participants between July 2013 and February 2017; meanwhile data collected from 8289 participants (34–80 years of age) between July 2013 and February 2016 were used for the CCT measurements.

All participants underwent ophthalmic examinations, including objective determinations of the corneal thicknesses (TX-20P; Canon, Tokyo, Japan) and axial lengths (IOLMaster; Carl Zeiss Meditec, Inc., Dublin, CA, USA). The study participants comprised individuals with available DNA samples; CCTs for both eyes; and information regarding the patient's age, sex, and axial length of both eyes. Of the participants, 84 who lacked CCT data for both eyes, 617 with a history of intraocular surgery, and 480 who lacked axial length data for both eyes were excluded. Of the remaining participants, samples from those who underwent genome-wide SNP genotyping were used for the discovery GWAS, and remaining subset of samples was used for the replication stage.

The Kyoto University Graduate School and Faculty of Medicine Ethics Committee and the Nagahama Municipal Review Board of Personal Information Protection approved the study protocol and the procedures used to obtain informed consent (Kyoto University Graduate School and Faculty of Medicine Ethics Committee approval number; G0278). All study procedures adhered to the tenets of the Declaration of Helsinki. All participants were fully informed of the purpose and procedures of the study, and written consent was obtained from each participant. Patient records and information were anonymised prior to analysis.

**Genome-wide SNP genotyping.** Genomic DNA was extracted from peripheral blood samples according to standard laboratory procedures. Genome-wide SNP genotyping was performed on samples from 5299 participants. A series of Bead-Chip DNA arrays, namely the HumanHap610 Quad (1818 samples), HumanOmni2.5-4 (1606 samples), HumanOmni2.5-8 (375 samples), HumanOmni2.5 s (671 samples), CoreExome24 (1,722 samples), and HumanExome (671 samples; Illumina, San Diego, CA, USA) arrays, were used for the analysis. Some samples were repeatedly genotyped using multiple kinds of DNA arrays.

Quality-control filters were used to remove poorly performing samples and SNP markers. SNPs with a call rate of <99%, an MAF of <1%, and a significant deviation from the Hardy–Weinberg equilibrium ($P < 1.0 \times 10^{-6}$) were excluded from further statistical analysis. Samples with a call rate of <95% were excluded from the analysis. Among the remaining participants, 636 were estimated to have a first- or second-degree kinship within this population (pi-hat > 0.35, PLINK ver. 1.90 [https://www.cog-genomics.org/plink/1.9/]) and were excluded from the analysis.

Genotype imputation was performed using MACH software (http://www.sph.umich.edu/csg/abecasis/MACH/tour/imputation.html), with a 1000 Genomes dataset (phase 3, v5 release) serving as a reference panel. Imputed SNPs for which $R^2 < 0.5$ were excluded from the association analysis. Finally, 4,710,779 SNPs from 3584 individuals were used for the discovery stage analysis.

**Replication genotyping for the GWAS of CCT.** To conduct a replication study for confirming genetic associations with CCT, genotypes of the remaining samples from the Nagahama Study ($n = 3422$) were determined using commercially available TaqMan SNP assays and the ABI PRISM 7700 system (Applied Biosystems, Foster City, CA, USA). Fourteen patients without CCT data in both eyes, six patients lacking axial length data for both eyes, and 460 patients with history of intraocular surgery were excluded. Finally, 2942 samples were used for the replication stage analysis.

**Trans-ethnic replication.** All patients enrolled in the study provided informed consent, and ethical approval was granted by Singapore Malay Eye Study (SiMES) group, the SingHealth Centralized Institutional Review Board (SCES), and the Singapore Indian Eye Study (SINDI) group. For the trans-ethnic replication stage, we included three Asian cohorts, namely the SiMES, SCES, and SINDI cohorts, which consisted of Malay ($n = 2510$), Chinese ($n = 2469$), and Indian ($n = 2508$) subjects, respectively. We also included publicly available data on a Latino population ($n = 3584$). Detailed information and the sample-collection methods for these cohorts can be found in the Supplementary Table 4 and Supplementary Note and in a previous publication[17].

**Statistics and reproducibility.** Genome-wide linear-regression analysis was conducted for the average CCT in both eyes in the discovery set of the Nagahama Cohort ($N = 3584$). This regression framework enabled us to adjust for covariates, such as age, sex, the average axial length in both eyes, and the first principal component. In the replication analysis with data from the SiMES ($N = 2510$), SCES ($N = 2469$), and SINDI ($N = 2508$) cohorts, we performed association testing under an additive model to examine the effect of the risk allele, while adjusting for the age, sex, and at least the first five principle components. Experiment-wide significance in the discovery stage was set at $P < 5.0 \times 10^{-8}$. All meta-analyses were performed using an inverse variance weighted fixed effect model. Thereafter, differences were considered statistically significant at $P < 0.05$. Deviations in genotype distributions from the Hardy–Weinberg equilibrium were assessed by performing Chi-squared tests. These statistical analyses were performed using R software (R Foundation for Statistical Computing, Vienna, Austria; available at http://www.rproject.org/, version 3.5.2) and PLINK software, version 1.90 (https://www.cog-genomics.org/plink/1.9/).

**Association of identified keratoconus-susceptibility gene with keratoconus.** To investigate the association between identified gene and keratoconus, we performed a case-control analysis using data from 179 patients with keratoconus from Yokohama City University and pooled the data of 9589 Japanese healthy control subjects from Yokohama City University, Nagahama Cohort and Tohoku Medical Megabank Project[30,31] A detailed description of these cohorts and the genotyping methods used are provided in the Supplementary Note. Briefly, both cases and controls were recruited from the Yokohama City University, and genotyped using the same genotyping platform, OmniExpress, and imputed using the same pipelines. Control samples from the Nagahama Cohort were genotyped as described above. Control samples from the Tohoku Medical Megabank Project were derived from a publicly available database. Differences in the genotype distributions of these cohorts were tested using the Cochran–Armitage test. $P < 0.05$ was set as the threshold of statistical significance.

**Immunohistochemical staining.** To confirm the expression of identified gene, we conducted immunohistochemical staining with the corneas of normal adult C57BL/6 mice. After each mouse was sacrificed, the eyes were enucleated and fixed with 4% paraformaldehyde in 0.1 M PB for 20 min. Expression in the cornea was examined in vertical sections. For this purpose, the eyes were cryoprotected in a sucrose gradient (10 and 30% w/v sucrose in 0.1 M PB), and 13-μm cryostat

sections were cut. The sections were blocked for 1 h in a solution containing 10% normal goat serum (Millipore), 0.3% Triton X-100 (Bio-Rad, Hercules, CA, USA), and 0.1 M PB. The sections were incubated overnight at 4 °C with a primary rabbit anti-human STON2 antibody (catalogue number NBP1-90658; Novus Biologicals, LLC), diluted 1:10 in blocking solution. Subsequently, the sections were incubated for 1 h with a secondary Alexa Fluor 488-conjugated goat anti-rabit IgG antibody (catalogue number A-11005; Invitrogen, Eugene, OR, USA). All incubation steps were performed at 4 °C. All images were obtained using a Keyence BZ-9000 microscope.

**Predictive pathway analysis using the WDD system**. WDD is one of the most famous cognitive-computing systems. The details of the WDD predictive analysis algorithm are described elsewhere[15]; thus, the system is introduced briefly below. WDD is configured to support life sciences research and extracts text features from medical literature to identify new connections between entities of interest, such as genes and diseases. From these documents, Watson can create semantic connections between a known set of genes associated with keratoconus and candidate genes. WDD can also rank the candidate genes by their degrees of similarity to the known set, using a graph-diffusion algorithm.

To detect additional keratoconus-susceptibility loci and infer pathways associated with keratoconus, we performed WDD predictive analysis. Eight previously reported keratoconus-susceptibility genes (*FNDC3B*, *COL5A1*, *FOXO1*, *MPDZ*, *NF1B*, *RXRA*, *BANP*, and *ZNF469*) and the newly identified keratoconus-susceptibility gene was input as a 'teaching gene'[12,32,33].

The candidate genes were selected as follows. First, from 1215 SNPs with a P-value of $<1.0 \times 10^{-4}$ in the discovery GWAS (Supplementary Data 1), we selected 745 SNPs that were located within a gene. These SNPs were distributed within 53 genomic regions. Second, genes that had already been input as 'teaching genes' and genes that had never been mentioned previously in the literature were automatically excluded, resulting in the exclusion of three and eight genes as candidates, respectively. Finally, 42 genes were selected as candidate genes. This process is summarised in Supplementary Table 2.

Based on the predictive analysis results of WDD, we selected possible keratoconus-susceptibility genes with respect to identified gene and assessed the associations between those genes and keratoconus. For association studies with keratoconus, we principally selected the top SNP within the genes. If genotype data of the SNP of interest was not available in any of the control cohorts, we employed proxy SNPs instead. An association study was performed using data from 179 patients with keratoconus and pooled data from 6651 healthy control subjects. The Cochran–Armitage test and Bonferroni's correction were applied for statistical assessments. The nominal P-value threshold was set to 0.05.

**Reporting summary**. Further information on research design is available in the Nature Research Reporting Summary linked to this article.

## Data availability
The complete GWAS summary data can be visualised at figshare (https://figshare.com/articles/dataset/CCT/12592529). The datasets generated during the current study are also available from the corresponding author on reasonable request.

## Code availability
IBM's Watson for Drug Discovery was used to select candidate genes in the present study (accessed on March 1st, 2018).

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

## Acknowledgements
This study was supported by a University Grant from Kyoto University; a Grant-in-Aid for Scientific Research from the Ministry of Education, Culture, Sports, Science & Technology in Japan; the Center of Innovation Program; the Global University Project from Japan Science and Technology Agency; the Practical Research Project for Rare/Intractable Diseases; Comprehensive Research on Aging and Health Science Research Grants for Dementia R&D from the Japan Agency for Medical Research and Development (AMED); a research grant from the Takeda Science Foundation; and SNEC HREF funds. We thank Ms. Hatsue Hamanaka and Ms. Miki Kokubo for assistance with genotyping. We are extremely grateful to the Nagahama City Office and the nonprofit organisation, Zeroji Club for their help in conducting the Nagahama Study. We also acknowledge the following sources of funding support for analysing the SiMES, SCES, and SINDI cohorts: the National Medical Research Council, Singapore (grant numbers NMRC/TCR/002-SERI/2008 (R626/47/2008TCR), CSA R613/34/2008, NMRC 0796/2003, and STaR/0003/2008), the National Research Foundation of Singapore, the Bio-medical Research Council, Singapore (grant numbers BMRC 09/1/35/19/616, 08/1/35/

19/550, and 10/1/35/19/675), and the Genome Institute of Singapore (grant number GIS/12-AR2105). The Singapore Tissue Network and the Genome Institute of Singapore (Agency for Science, Technology and Research, Singapore) provided services.

## Author contributions

Y.H. and M.M. designed the study. M.M., S.M.S., and K.Y. supervised the design of the study. Y.H., M.M., A.M., S.I., M.Y., C.C.K., S.M.S., and K.Y. analysed the data. M.M., R.Y., and C.Y.C. supervised the analysis. Y.H., M.M., Y.T., R.Y., F.M., and the Nagahama Study Group managed the cohort data. Y.H., M.M., A.M., S.I., N.U.A., E.N., Y.M., H.N., C.C.K., S.S.M., R.Y., C.C., N.M., A.T., and K.Y. gathered and analysed the clinical data. Y.H. and M.M. drafted the paper. All authors critically revised and gave final approval to this manuscript.

## Competing interests

The authors declare no competing interests.

## Additional information

## The Nagahama Study Group

Yasuharu Tabara[2], Takahisa Kawaguchi[2], Kazuya Setoh[2], Fumihiko Matsuda[2], Yoshimitsu Takahashi[9], Takeo Nakayama[9] & Shinji Kosugi[10]

[9]Department of Health Informatics, Kyoto University School of Public Health, Kyoto, Japan. [10]Department of Medical Ethics and Medical Genetics, Kyoto University School of Public Health, Kyoto, Japan.

