## [Peer Review File · Communications Biology]

Reviewers' comments:

Reviewer #1 (Remarks to the Author):

CCT study. In their research, they combined two-stage GWAS and IBM's Watson question-answering computer system.

It is an interesting combination of methods and gives a chance to use such combinations also in the study of other diseases.

The association analysis for CCT was carried out in the Japanese population. Moreover, what is worth emphasizing that the authors have confirmed the presented results of the association analysis in other populations.

I find the manuscript interesting and well written, but I have a few comments.

1) As a result of the research, the authors indicate two new loci: STON2 and SMAD3. However, I cannot entirely agree with the use of the "novel" term for the SMAD3 gene because its expression has already been repeatedly tested in the context of the keratoconus, e.g., Priyadarsini et al. 2015, Sharif et al. 2017.

2) The methods section should include the exact consent numbers of bioethics committees.

3) Authors refer that: "Eight previously reported keratoconus-susceptibility genes (FNDC3B, COL5A1, FOXO1, MPDZ, NF1B, 204 RXRA, BANP, and ZFF460) and the newly identified gene were input as 'teaching genes'". However, I was not able to find what was "newly identified gene". I think that a table with a full training set should be included in supplementary.

Reviewer #2 (Remarks to the Author):

This manuscript describes a GWAS approach & a functional validation in a mouse model & predictive analysis using machine learning to identify new potential target loci for keratoconus development. The manuscript is well written and identifies an intriguing approach to find new pathways in relatively rare diseases.

Major comments:

Three methods have been combined in one paper (GWAS, mouse model, computational analysis). The GWAS outcomes appear solid and well executed. The translation to keratoconus is somewhat more problematic based on a relatively small keratoconus share of data (179) especially regarding the massive healthy databases that have been entered. Furthermore, the mouse model to me is not that convincing, since it's not even a disease KC model. This should preferably be replicated in human KC tissue. Finally, the computational analysis appear an independent part of this research project. In other word, are 3 papers merged into one? The proposed approach in making GWAS more efficient is intriguing though, but if that's the scope of the manuscript, maybe the authors can contemplate about writing a pure methodological manuscript.

Specific comments:

p4l44: the incidence of keratoconus appears much higher than this outdated reference suggests. Please see Godefrooij et al in Am J Ophthalmology 2017.

p4l45: these references are really outdated as well and stem from a pre-CXL era. A better reference is <https://www.ncbi.nlm.nih.gov/pubmed/27336399>. A nuance is further that corneal transplantation rates for KC dropped recently, see Godefrooij et al in Acta Ophthalmologica 2016 and

The previous work regarding GWAS in keratoconus and CCT is adequately described in the introduction. The interesting fact that previous GWAS outcomes could not be translated in a functional model could be elucidated further. See the review: <https://www.ncbi.nlm.nih.gov/pubmed/29111844>

p5173 I'm a bit puzzled that these results are mentioned in the introduction. It appears as if these outcomes were derived from prior studies (quod non). I'd advice to rewrite this section.

p5188: how could you exclude latent or form fruste keratoconus in this assumed healthy population? Little data is available to excluded keratoconus, e.g. refractive state, corneal topography (see the DUCK score, <https://www.ncbi.nlm.nih.gov/pubmed/30920597>). Can you contamplate how this effect might be mitigated?

p5186: what is meant by the 'second stage' of the study, and why where only individuals >34 selected? Keratoconus is a disease with a peek incidence in adolescence. This must be a deliberate choice by the authors, please elucidate. Why were only residents without physical impairments included? keratoconus is associated with a myriad of systemic diseases and Trisomy 21. this selection bias might preclude the identification of potential pathways.

p6193: did you consider multiple imputation, if missings can be assumed at random?

p6197: is the replication the same as the second stage?

p81142: albeit practically infeasible it would have been nice to include more caucasian and arabic/middle east samples, based on the geographical/racial difference in KC epidemiology and eye anatomy.

p81145-47: this was already mentioned earlier

p91166: To me it's somewhat unclear; are only 179 keratoconus patients included in the association part of the study? That's rather peculiar, considering all the efforts the authors have put in assembling prior databases in healthy individuals. These are presumably all Japanese individuals as well.

P101203: I really like this approach, where earlier GWAS outcomes are entered in the WDD. Therefore a myriad of solid loci are entered in the computational analysis. A Manhattan figure is given in fig1, this represents their GWAS. I would find a plot on the genes that entered computational analysis of added value (or maybe figure 1 can be enriched with this information)

p121234: clear remark on FNDC3B. How where other previously identified SNPs reported (ie OL5A1, FOXO1, MPDZ, NF1B, RXRA, BANP, and ZFF460) ?

p121245: how to put this replication in context? Can you elucidate why STON2 wasn't picked up before in these GWAS?

P131270: this translational step appears to me as a keypoint in this publication (this is added in methodology over previous GWAS). It merits more attention and backing by external studies. Could you integrate the data from Li et al from their KC GWAS?

p141282: please elucidate whether STON2 is expressed (or absent) in eye/brain tissue or collagen rich tissues. For me, there are too little functional clues that STON2 actual plays a role in KC development based on this mouse model.

p14|289: great execution of the WDD system, but could you explain the choices made in the selection of the stream? How now can we be certain that the first part of the study led to this final part, how are they connected? The WDD application appears an independent part of this study.

p15|312: I'm convinced by the GWAS strategy, by not convinced the mouse model, and since the relationship between the GWAS outcomes <-> non-KC mouse model <-> computational analysis has several assumptions, and a small N of KC cases, I'm puzzled. The presence of STON2 in a mouse model is not directly linked to a human-KC model, or human-KC samples (acquired after corneal transplant surgery for instance).

The WDD analysis to me appears as an independent method to identify pathways, and it's relationship with the authors' GWAS is unclear to me.

p15|320: this is indeed a major strength.

p16|335: these downstream effects could be attributed to other cases as well, most notably physical damage by eye-rubbing, or immunological changes (please refer to <https://www.ncbi.nlm.nih.gov/pubmed/26235733> in this aspect). The current line of reasoning is to direct for me.

p18|385: here the conclusion is framed much more nuanced and to -the -point.

Reviewer #3 (Remarks to the Author):

This article reports the results of a GWAS for central corneal thickness (CCT) in a sample of 3584 healthy Japanese volunteers (Nagahama cohort). Of the two genome-wide significant loci identified, one was already known to be associated with CCT (FNDC3B) while the other was novel (STON2; lead SNP rs2371597). Association of STON2 SNP rs2371597 with CCT was replicated in cohorts of Malay, Chinese, and Indian ethnicity recruited from Singapore. This variant was also found to be associated with keratoconus in a case-control sample from Japan (179 cases, 11084 controls; OR=1.27, P=0.04) and shown to be an eQTL for STON2 in some GTEx tissues. STON2 was expressed in basal corneal epithelial cells of mouse cornea. Furthermore, a bioinformatics literature mining analysis using the IBM Watson Drug Discovery algorithm identified SMAD3 as an additional candidate gene for keratoconus. In support of this theory, SNP rs12912547 in SMAD3 was associated with keratoconus in the case-control sample (R=1.44, P=0.001).

In general these findings are interesting and have both fundamental mechanistic relevance and clinical relevance. The GWAS component of the manuscript is scientifically convincing, whereas more clarification is required to demonstrate the validity of the Watson Drug Discovery analysis. These points are covered in more detail below.

Specific points

1. L52. The recent GWAS for keratoconus by Hardcastle et al. ARVO abstract # 4249 (2019) should be cited.
2. L133. Were any of the 2942 participants in the Nagahama replication sample related to any of the 3584 participants in the Nagahama discovery GWAS sample? If so, then this is not independent

replication; the related individuals should be excluded.

3. L154. It is unusual to include only 1 PC as a GWAS covariate; typically 5, 10 or 20 PCs are included. Please justify this a priori choice of 1 PC and report if the 2 genome-wide significant associations from the discovery GWAS were altered if adjustment was made for 10 or 20 PCs.

4. L158. Give details of the fixed effects meta-analysis weighting, e.g. inverse variance.

5. L171. State explicitly if the cases and controls were genotyped using the same genotyping platform and if the imputation pipelines used for cases and controls were exactly the same. If cases and controls were genotyped separately, this can lead to spurious association signals. How did the authors assess whether associations were truly due to disease status and not simply a technical artefact?

6. L171. Which covariates were included in the case-control GWAS? Were any participants related?

7. L204. Cite references for the studies describing these known risk SNPs.

8. L217. More detail is required on the assessment of candidate genes identified by the WDD analysis. Please report (a) how many genes in total were identified in the WDD analysis, (b) were all of these genes assessed in the keratoconus case-control sample? (If only genes in the FOXO1-SMAD stream were assessed this may appear to be cherry-picking) (c) was 1 SNP examined per gene? (d) was the reported p-value of $p=0.001$ corrected for multiple testing?

9. L281. Does an eQTL database exist for cornea or corneal epithelium? If so, please report results for the lead STON2 SNP or a surrogate. If not, please mention this lack of a suitably matched eQTL database in the text.

10. L349. Test for shared molecular mechanisms between keratoconus and schizophrenia using LD score regression.

11. Table 1. Instead of the "Nearby gene" column, it would be helpful to state whether the SNP in genic, intronic, etc.

12. Table 3 could be omitted and the salient information presented in the text.

13. Figure 1 and 2. The small font size made the text difficult to read.

14. Figure 3 legend. Define blue label for nuclei.

15. Figure 5. Define symbols.

16. General. Please present 95% CI when reporting odds ratios.

Grammatical and typographical suggestions

1. Abstract L33. Use "heritable" in place of "inheritable".

2. Abstract L34. I suggest, "...we identified a novel locus for CCT, STON2 rs2371597..."

3. Abstract L35. Correctly format of p-value.

4. L142. The meaning of "one open data" is not clear. Please re-word.

5. L157. I suggest, "...experiment-wide significance..."

6. L286. What is meant by "remarkable expression". Please re-word.

7. L302. Define "ToMMo".

Reviewers' comments:

Reviewer #1

Reviewer's comment
CCT study. In their research, they combined two-stage GWAS and IBM's Watson question-answering computer system. It is an interesting combination of methods and gives a chance to use such combinations also in the study of other diseases. The association analysis for CCT was carried out in the Japanese population. Moreover, what is worth emphasizing that the authors have confirmed the presented results of the association analysis in other populations.
Response to Reviewer
Thank you for your comments. We believe the current method can be applied to other studies as well.
Changes in the Manuscript
-

Reviewer's comment
I find the manuscript interesting and well written, but I have a few comments. 1) As a result of the research, the authors indicate two new loci: STON2 and SMAD3. However, I cannot entirely agree with the use of the "novel" term for the SMAD3 gene because its expression has already been repeatedly tested in the context of the keratoconus, e.g., Priyadarsini et al. 2015, Sharif et al. 2017.
Response to Reviewer
Thank you for your comment. Although the importance of SMADs, including SMAD3, in keratoconus pathogenesis was previously evaluated, the association of SNPs in SMAD3 with keratoconus development was not confirmed previously by genetic studies. While we completely agree with the reviewer's comment, in the area of genetic studies, a susceptibility gene whose association with the disease has not been previously reported is generally referred to as "a novel susceptibility gene" regardless of the previous molecular-biological results. Thus, in this study, we would like to respectfully retain the term "novel". Thank you for your understanding.
Changes in the Manuscript
-

Reviewer's comment
2) The methods section should include the exact consent numbers of bioethics committees.
Response to Reviewer
Thank you for your comment. For the initial follow-up of the Nagahama cohort, consent was obtained from 9,850 participants (July 2013 to February 2017). However, since CCT measurement was not performed after February 2016, we used the data collected from 8,289 participants (July 2013 and February 2016) in the current study. We revised the manuscript as follows. We also provided the approval number that was obtained from Kyoto University Graduate School and Faculty of Medicine Ethics Committee.
Changes in the Manuscript
For the two-staged GWAS, we analysed data from healthy Japanese volunteers enrolled in the Nagahama Prospective Cohort for Comprehensive Human Bioscience (the Nagahama Study). The initial follow-up data were collected from 9,850 participants between July 2013 and February 2017; meanwhile data collected from 8,289 participants (34 – 80 years of age) between July 2013 and February 2016 were used for the CCT measurements. (Page 13, line 275-280) The Kyoto University Graduate School and Faculty of Medicine Ethics Committee and the Nagahama Municipal Review Board of Personal Information Protection approved the study protocol and the procedures used to obtain informed consent (Kyoto University Graduate School and Faculty of Medicine Ethics Committee approval number; G0278). All study procedures adhered to the tenets of the Declaration of Helsinki. All participants were fully informed of the purpose and procedures of the study, and written consent was obtained from each participant. Patient records and information were anonymised prior to analysis. (Page 14, line 291-298)

Reviewer's comment
3) Authors refer that: "Eight previously reported keratoconus-susceptibility genes (FNDC3B , COL5A1 , FOXO1 , MPDZ , NF1B , 204 RXRA , BANP , and ZFF460) and the newly identified gene were input as 'teaching genes'". However, I was not able to find what was "newly identified gene". I think that a table with a full training set should be included in supplementary.
Response to Reviewer
We apologize for our poor initial explanation. We included STON2 as the newly identified keratoconus susceptibility gene in WDD analysis. In addition, "newly identified gene" is not accurate; we should have instead used "newly identified susceptibility gene". We also revised the misspelling of the gene name, from " ZFF460 " to " ZNF469 " throughout the manuscript. We clarified it in the Methods and Result section as follows.
Changes in the Manuscript
To detect additional keratoconus-susceptibility loci and infer pathways associated with keratoconus, we performed WDD predictive analysis. Eight previously reported keratoconus-susceptibility genes (FNDC3B , COL5A1 , FOXO1 , MPDZ , NF1B , RXRA , BANP , and ZNF469) and the newly identified keratoconus susceptibility gene was input as a 'teaching gene'. ^{12,20,21} (Page 18, line 396-400)

Reviewer #2

Reviewer's comment
This manuscript describes a GWAS approach & a functional validation in a mouse model & predictive analysis using machine learning to identify new potential target loci for keratoconus development. The manuscript is well written and identifies an intriguing approach to find new pathways in relatively rare diseases.
Response to Reviewer
Thank you for your comments. We believe the current approach shows a novel application of artificial intelligence.
Changes in the Manuscript
-

Reviewer's comment
Major comments: Three methods have been combined in one paper (GWAS, mouse model, computational analysis). The GWAS outcomes appear solid and well executed.
Response to Reviewer
Thank you for your comments.
Changes in the Manuscript
-

Reviewer's comment
The translation to keratoconus is somewhat more problematic based on a relatively small keratoconus share of data (179) especially regarding the massive healthy databases that have been entered.
Response to Reviewer
Thank you for your comment. We agree with the reviewer that the sample size of keratoconus is relatively small. However, since keratoconus is a rare disease, we were unable to recruit additional cases for this study. Nevertheless, to increase the statistical power of case-control analysis, we recruited as many controls as possible using publicly available database. As a result, we could estimate population allele frequency of healthy individuals precisely enough to detect statistically-significant differences, despite the imprecise estimation of population allele frequency of keratoconus individuals due to small sample size. Though we understand the reviewer's concern, we believe this is a useful statistical method to circumvent the issue. However, in our future study, we will collect additional keratoconus samples to achieve higher statistical power.
Changes in the Manuscript
-

Reviewer's comment
Furthermore, the mouse model to me is not that convincing, since it's not even a disease KC model. This should preferably be replicated in human KC tissue.
Response to Reviewer
Thank you for your comments. In the immunostaining section, we only confirmed the expression of STON2 in mouse corneal tissue and we agree that the evaluation of gene expression in human KC tissue will be required to reveal a more detailed role of STON2 in keratoconus development. However, since the expression of STON2 has never been evaluated in human KC tissue, KC mouse model or healthy mouse cornea, we believe the confirmation of STON2 expression in healthy mouse cornea can be a first step towards further immunohistological evaluation of STON2. This reviewer's comment would also be our future challenges. Thank you for your important suggestion.
Changes in the Manuscript
-

Reviewer's comment
Finally, the computational analysis appear an independent part of this research project. In other word, are 3 papers merged into one? The proposed approach in making GWAS more efficient is intriguing though, but if that's the scope of the manuscript, maybe the authors can contemplate about writing a pure methodological manuscript.
Response to Reviewer
Thank you for your comments and suggestion. As the reviewer pointed out, the approach to making GWAS more efficient is one of the appealing aspects of the current study. However, since we did not apply the approach to other GWASs, we could not demonstrate its generalizability. Nevertheless, we believe it can be generalized. This is one of the reasons why we did not write a pure methodological manuscript. Another reason is that we are more interested in reporting the novel susceptibility genes for keratoconus, rather than pursuing the methodology. Furthermore, although the reviewer stated that the computational analysis appeared to be an independent aspect of this research project, we believe they are all connected. Specifically, in the WDD section, we sought to evaluate the functional connection between a novel keratoconus susceptibility gene STON2 and eight established KC susceptibility genes by evaluating the genes harnessing them (FOXO1-SMAD-STON2 stream). In other words, if we had not identified STON2 as a novel susceptibility gene for keratoconus, we would not have focused on the stream, and would not have identified SMAD3 as a novel susceptibility gene for keratoconus.
Changes in the Manuscript
-

Reviewer's comment
Specific comments: p4l44: the incidence of keratoconus appears much higher than this outdated reference suggests. Please see Godefrooij et al in Am J Ophthalmology 2017.
Response to Reviewer
Thank you for your advice. We agree that the original reference was out of data and that the more recent studies have reported higher prevalence. As such, we have cited the study that you have mentioned, in the revised manuscript.
Changes in the Manuscript
Keratoconus is a common, bilateral, noninflammatory type of corneal degeneration, affecting 1 out of every 375 people in the general population¹, and is a major indication for corneal transplantation in developed countries.²⁻⁴ (page 4, line 43-45)

Reviewer's comment
p4145: these references are really outdated as well and stem from a pre-CXL era. A better reference is https://www.ncbi.nlm.nih.gov/pubmed/27336399 . A nuance is further that corneal transplantation rates for KC dropped recently, see Godefrooij et al in Acta Ophthalmologica 2016 and
Response to Reviewer
Thank you for the updates. We carefully read the papers and revised the manuscript to include these references.
Changes in the Manuscript
Keratoconus is a common, bilateral, noninflammatory type of corneal degeneration, affecting 1 out of every 375 people in the general population ¹ , and is a major indication for corneal transplantation in developed countries. ²⁻⁴ (page 4, line 43-45)

Reviewer's comment
The previous work regarding GWAS in keratoconus and CCT is adequately described in the introduction. The interesting fact that previous GWAS outcomes could not be translated in a functional model could be elucidated further. See the review: https://www.ncbi.nlm.nih.gov/pubmed/29111844
Response to Reviewer
Thank you for sharing this additional information. As mentioned in the review, it may be important to identify variants with smaller effect sizes, which may lead to further progress in KC research. Our approach using WDD may provide a novel viewpoint and approach which enables us to detect KC susceptibility genes that could not be identified through simple large-sized GWAS. We cited the review and revised the manuscript as follows.
Changes in the Manuscript
In addition, pathway analysis of keratoconus has not been performed because of the lack of reliable GWASs of keratoconus. To reveal the specific role of various genes in keratoconus development, it may be important to identify susceptibility genes with relatively smaller effect size which cannot be identified through large sample size GWAS studies.¹⁴ (page 4-5, line 64-69)

Reviewer's comment
p5173 I'm a bit puzzled that these results are mentioned in the introduction. IT appears as if these outcomes were derived from prior studies (quod non). I'd advice to rewrite this section.
Response to Reviewer
As the reviewer commented, the results of the current study were mentioned in this section. We revised the section to improve readability and flow. Thank you for this suggestion.
Changes in the Manuscript
In this study, we used the latest artificial-intelligence technology, IBM's Watson for Drug Discovery (WDD), to select candidate genes from the results of our newly performed GWAS on CCT. WDD, which has been configured to support life science research, can help in understanding and identifying relationships between molecules/genes, using various types of enormous datasets from structured databases and PubMed database. ^{15,16} To identify new keratoconus susceptibility loci, we performed a traditional two-stage GWAS on CCT using a Japanese community-based cohort, and assessed the association of CCT susceptibility loci with keratoconus occurrence. Further, we assessed new keratoconus susceptibility genes derived from the combination of GWAS results and WDD. (page 5, line 70-79)

Reviewer's comment
p5188: how could you exclude latent or form fruste keratoconus in this assumed healthy population? Little data is available to excluded keratoconus, e.g. refractive state, corneal topography (see the DUCK score, https://www.ncbi.nlm.nih.gov/pubmed/30920597). Can you contemplate how this effect might be mitigated?
Response to Reviewer
Thank you for your comments. In fact, we did not exclude keratoconus patients from the healthy population in the current study, so you are correct in stating that it is possible that a few keratoconus patients were included in the healthy subjects. We agree that excluding keratoconus patients from controls would be a more ideal approach. However, since keratoconus is a relatively rare disease, the effect of the contamination on the genetic association study seems to be minimal. In addition, cohort studies generally do not have detailed clinical data. For instance, the Nagahama Study did not have corneal shape analysis data. This is why the current approach is widely accepted for genetic association study of rare diseases.¹⁻³ To address the reviewer's concern, we performed GWAS on corneal thickness after excluding samples with high astigmatism (> -2 D) to reduce the effect of unintentionally included keratoconus patients in the discovery GWAS. We found that rs2371597 in STON2 still showed genome wide significance level of association with corneal thickness ($\beta = 5.51$, $P = 3.63 \times 10^{-11}$). Although keratoconus patients were not excluded from the discovery GWAS, we believe the result of this study is robust. References  1. Fritsche LG, Chen W, Schu M, et al. Seven new loci associated with age-related macular degeneration. Nat Genet 2013;45:433–439. 2. Hysi PG, Cheng CY, Springelkamp H, et al. Genome-wide analysis of multi-ancestry cohorts identifies new loci influencing intraocular pressure and susceptibility to glaucoma. Nat Genet 2014;46:1126–1130. 3. Arakawa S, Takahashi A, Ashikawa K, et al. Genome-wide association study identifies two susceptibility loci for exudative age-related macular degeneration in the Japanese population. Nat Genet 2011;43:1001–1005.
Changes in the Manuscript
-

Reviewer's comment
p5186: what is meant by the 'second stage' of the study, and why were only individuals >34 selected? Keratoconus is a disease with a peak incidence in adolescence. This must be a deliberate choice by the authors, please elucidate. Why were only residents without physical impairments included? keratoconus is associated with a myriad of systemic diseases and Trisomy 21. this selection bias might preclude the identification of potential pathways.
Response to Reviewer
We apologize for not describing this more clearly in the original manuscript. In the current study, we used a community-based healthy Japanese cohort, the Nagahama Study,^{1,2} in which the healthy individuals (i.e. individuals without apparent impairment) aged 34 to 80 years voluntarily participated. They were divided into two groups; the individuals that had undergone genomic scanning were used for the first (i.e. discovery) stage, and the remainder of the individual were used for the second (i.e. replication) stage. In the second stage, the genotype distribution was determined using Taqman SNP genotyping assay. We then identified CCT-associated loci rs2371597 in STON2. To evaluate the association of this candidate SNP (rs2371597 in STON2) with keratoconus development, we recruited hospital-based keratoconus patients regardless of age, sex or systemic diseases. We did not exclude adolescent patients or patients with physical impairment at this stage. References  1. Narahara, M. et al. Large-scale East-Asian eQTL mapping reveals novel candidate genes for LD mapping and the genomic landscape of transcriptional effects of sequence variants. PLoS One 9, e100924 (2014). 2. Higasa, K. et al. Human genetic variation database, a reference database of genetic variations in the Japanese population. J. Hum. Genet. 61, 547-553 (2016).
Changes in the Manuscript
-

Reviewer's comment
p6l93: did you consider multiple imputation, if missings can be assumed at random?
Response to Reviewer
Thank you for the comment. We think the missing occurred completely at random (MCAR) or at random (MAR), so that multiple imputation can work. However, we did not consider it in the current study.
Changes in the Manuscript
-

Reviewer's comment
p6197: is the replication the same as the second stage?
Response to Reviewer
Yes. The replication stage is the same as "the second stage".
Changes in the Manuscript
-

Reviewer's comment
p8l142: albeit practically infeasible it would have been nice to include more Caucasian and Arabic/middle east samples, based on the geographical/racial difference in KC epidemiology and eye anatomy.
Response to Reviewer
Thank you for your comment. As you mentioned, it may provide more interesting knowledge to perform transethnic comparisons. We want to consider this for future studies.
Changes in the Manuscript
-

Reviewer's comment
p81145-47: this was already mentioned earlier
Response to Reviewer
We removed the sentence.
Changes in the Manuscript
Detailed information and the sample-collection methods for these cohorts can be found in the Supplementary Table 6 and Supplementary Note and in a previous publication. ¹⁷ Genomic DNA was extracted from peripheral blood samples according to standard laboratory procedures. Individuals who had undergone ocular surgery or ocular laser treatment were excluded from the analysis. (page 16, line 338-340)

Reviewer's comment
p91166: To me it's somewhat unclear; are only 179 keratoconus patients included in the association part of the study? That's rather peculiar, considering all the efforts the authors have put in assembling prior databases in healthy individuals. These are presumably all Japanese individuals as well.
Response to Reviewer
We agree that the number of cases is relatively small. However, it is often the case that DNA samples of healthy individuals are collected relatively easily through a cohort study, while those from diseased individuals cannot. For example, although the previous meta-QTL of CCT included 25,910 individuals for CCT QTL, it included only 933 patients with keratoconus.¹ This is comparable to the current study which included 3,584 individuals for CCT and 179 patients with keratoconus ($933/25910 = 0.036$, $179/3584 = 0.050$). Thus, although we agree with the reviewer that including more keratoconus samples will enhance the power of this study, we believe the current sample size was acceptable considering the relatively low prevalence of keratoconus. Moreover, we were able to identify two novel keratoconus susceptibility genes through the combination of GWAS and WDD. References 1. Iglesias, A. I. et al. Cross-ancestry genome-wide association analysis of corneal thickness strengthens link between complex and Mendelian eye diseases. Nat. Commun. 9, 1864 (2018).
Changes in the Manuscript
-

Reviewer's comment

P101203: I really like this approach, where earlier GWAS outcomes are entered in the WDD. Therefore a myriad of solid loci are entered in the computational analysis. A Manhattan figure is given in fig1, this represents their GWAS. I would find a plot on the genes that entered computational analysis of added value (or maybe figure 1 can be enriched with this information)
]

Response to Reviewer

Thank you for your comment. We provided the threshold line of $P = 1.0 \times 10^{-4}$, which is used to pick up candidate genes for WDD in Figure 1 and revised the legend.

Changes in the Manuscript

Figure 1. Manhattan plot of QTL data from our GWAS of CCT.

The plot shows $-\log_{10}$ -transformed P values for all SNPs adjusted for age, sex, the average of axial length in both eyes, and the first principal component. The horizontal line represents the genome-wide significance threshold of 5.0×10^{-8} , and the lower dashed line represents the threshold of 1.0×10^{-4} to identify candidate genes for Watson Drug Discovery analysis.

Reviewer's comment
p12l234: clear remark on FNDC3B . How where other previously identified SNPs reported (ie OL5A1, FOXO1, MPDZ, NF1B, RXRA, BANP, and ZFF460) ?
Response to Reviewer
Thank you for the question. Since FNDC3B exceeded the genome wide significance level of association, we specifically mentioned it in this section. The association of other previously reported SNPs are summarized in the later section (page 13, line 259-270) and Supplementary Table 3; they also demonstrated mild associations with CCT with the same direction.
Changes in the Manuscript
-

Reviewer's comment
p12l245: how to put this replication in context? Can you elucidate why STON2 wasn't picked up before in these GWAS?
Response to Reviewer
Although the associations between STON2 and SMAD3 and keratoconus are novel findings of the current study, the association between STON2 with CCT was already reported in the previous CCT GWAS, which was published when we were drafting the manuscript.¹ In contrast to the previous study including 25,000 samples, we identified the contribution of STON2 on CCT using relatively small samples. We speculate that the effect of STON2 on CCT may be stronger in Japanese than that in other ethnicities. References 1. Iglesias, A. I. et al. Cross-ancestry genome-wide association analysis of corneal thickness strengthens link between complex and Mendelian eye diseases. Nat. Commun. 9, 1864 (2018).
Changes in the Manuscript
-

Reviewer's comment
P131270: this translational step appears to me as a keypoint in this publication (this is added in methodology over previous GWAS). It merits more attention and backing by external studies. Could you integrate the data from Li et al from their KC GWAS?
Response to Reviewer
Thank you for the suggestion. Accordingly, we attempted to contact Li et al, and found that their data were included in the CCT meta-QTL paper.¹ In their Caucasian data, the proxy SNP of rs2371597 (rs56223983, $R^2 = 0.71$ in East Asian, using LDlink) was not significantly associated with keratoconus ($P = 0.87$). However, we believe this result does not deteriorate the current result since such ethnic heterogeneity is sometimes observed in the genetic study. As the reviewer would be familiar with, the prevalence of keratoconus is reported to be different between ethnicities. For example, a higher incidence in Asians compared to Caucasians at a ratio of 7.5:1, was reported.² In an Israeli epidemiological study, keratoconus prevalence was reported to be as high as 2.3%.³ Moreover, increased incidence of keratoconus was reported in an Indian rural population and a Saudi Arabian population with 2.3 and 20 cases per 100,000 respectively.^{4,5} We speculate that the genetic and pathological background of keratoconus can differ between Asian and Caucasian, and that some keratoconus susceptibility genes may be identified in Asian that are not associated with keratoconus development in Caucasians. Further studies to compare the effect size of keratoconus susceptibility genes on keratoconus development across various ethnicities will serve to reveal these ethnical differences. We address these points in the revised Discussion section. References  1. Iglesias AI, Mishra A, Vitart V, et al. Cross-ancestry genome-wide association analysis of corneal thickness strengthens link between complex and Mendelian eye diseases. Nat Commun 2018;9:1864. [PMID:29760442] 2. Kok YO, Tan GF, Loon SC. Review: Keratoconus in Asia. Cornea. 2012; 31:581–593. [PubMed:22314815] 3. Millodot M, Shneor E, Albou S, Atlani E, Gordon-Shaag A. Prevalence and associated factors of keratoconus in Jerusalem: a cross-sectional study. Ophthalmic Epidemiol. 2011; 18:91–97.[PubMed: 21401417]

4. Jonas JB, Nangia V, Matin A, Kulkarni M, Bhojwani K. Prevalence and associations of keratoconus in rural maharashtra in central India: the central India eye and medical study. *Am J Ophthalmol.* 2009; 148:760–765. [PubMed: 19674732]
5. Assiri AA, Yousuf BI, Quantock AJ, Murphy PJ. Incidence and severity of keratoconus in Asirprovince, Saudi Arabia. *Br J Ophthalmol.* 2005; 89:1403–1406. [PubMed: 16234439]

Changes in the Manuscript

We also identified STON2 rs2371597 (located at chromosome14q31) as a novel keratoconus-susceptibility SNP. STON2 was previously identified as an endocytic adaptor dedicated to the retrieval of surface-stranded synaptic vesicle proteins.²² A previous genetic study reported that *STON2* SNPs were associated with CCT, but not with keratoconus development in Caucasians. Therefore, as the prevalence of keratoconus is known to be higher in the Asian population compared to Caucasians, we speculate that the genetic and pathological background of keratoconus differs between ethnicities. Moreover, as it is possible that certain keratoconus-susceptibility genes identified in Asians are not associated with keratoconus development in Caucasians, further studies are required to compare the effect size of keratoconus-susceptibility genes on keratoconus development across various ethnicities, which will serve to reveal ethnical differences.

The eQTL data revealed that the effect size of rs2371597 on *STON2* expression was strongest in skeletal muscle, which is rich in collagen, a protein that reportedly plays a key role in keratoconus pathogenesis. Although no previous reports have investigated the expression of *STON2* in human corneal tissue, we speculate that *STON2* may play an important role in keratoconus development by interacting with extracellular matrix remodelling. Our immunohistochemical staining results for *STON2* are compatible with a previous histopathological study of keratoconus, which showed that thinning of the corneal stroma, breaks in Bowman’s layer, and degeneration of the corneal epithelium were the characteristics of corneas in patients with keratoconus.^{2,23} *STON2* might be associated with the vulnerability to physical damage or immunological changes. Although biological studies examining the role of *STON2* in human corneal tissue are required to prove our hypothesis, pathways involving *STON2* may serve as novel targets for treating keratoconus by controlling basal cell degeneration.

(page 10-11, line 192-217)

Reviewer's comment
p14l282: please elucidate whether STON2 is expressed (or absent) in eye/brain tissue or collagen rich tissues. For me, there are too little functional clues that STON2 actually plays a role in KC development based on this mouse model.
Response to Reviewer
We appreciate the reviewer for this important suggestion. We accessed the GTEx portal again (accessed on 06 April, 2020), and found that the normalized effect size (NES) of rs2371597 on STON2 expression was strongest in the skeletal muscle (NES = -0.189, P = 1.3×10^{-8}) which is rich in collagen. Although there was no previous report on the expression of STON2 in human corneal tissue, the rs2371597 genotype may be associated with expression of STON2 in cornea considering both skeletal muscle and cornea are collagen rich tissues. STON2 might, therefore, play an important role in keratoconus development by interacting with collagen remodeling. We revised the Results and Discussion sections as follows.
Changes in the Manuscript
Expression of STON2 in human tissues and in the mouse cornea A search of a publicly available expression quantitative trait loci analysis (eQTL) database revealed that rs2371597 was significantly associated with STON2 expression (GTEx Portal; https://gtexportal.org/home/). A multi-tissue eQTL plot revealed that the normalized effect size (NES) of rs2371597 on STON2 expression was strongest in the skeletal muscle (NES = -0.189, P = 1.3×10^{-8}; Figure 2, https://www.gtexportal.org/home/snp/rs2371597) in which collagen plays an important role in providing its tensile strength and elasticity. However, data on the association of rs2371597 with gene expression in human corneal tissue was not available. Our immunohistochemical study of mouse corneas showed that STON2 was expressed in the corneal epithelial cell layer (Figure 2). Our results demonstrated that strong STON2 expression mainly occurred in basal cells rather than superficial cells in the corneal epithelium. In the stroma and endothelium layer, only minimal STON2 expression was observed. (page 7-8, line 137-149) The eQTL data revealed that the effect size of rs2371597 on STON2 expression was strongest in skeletal muscle, which is rich in collagen, a protein that reportedly plays a key role in keratoconus pathogenesis. Although no previous reports have investigated the expression of

STON2 in human corneal tissue, we speculate that *STON2* may play an important role in keratoconus development by interacting with extracellular matrix remodelling. Our immunohistochemical staining results for *STON2* are compatible with a previous histopathological study of keratoconus, which showed that thinning of the corneal stroma, breaks in Bowman's layer, and degeneration of the corneal epithelium were the characteristics of corneas in patients with keratoconus.^{2,23} *STON2* might be associated with the vulnerability to physical damage or immunological changes. Although biological studies examining the role of *STON2* in human corneal tissue are required to prove our hypothesis, pathways involving *STON2* may serve as novel targets for treating keratoconus by controlling basal cell degeneration.

(page 10-11, line 204-217)

Reviewer's comment
p14l289: great execution of the WDD system, but could you explain the choices made in the selection of the stream? How now can we be certain that the first part of the study led to this final part, how are they connected? The WDD application appears an independent part of this study.
Response to Reviewer
Thank you for your comment. In the first GWAS part, we identified STON2 as a novel keratoconus susceptibility gene. In the third WDD part, we included not only previously reported susceptibility genes but also STON2 as a “teacher gene”. Then, among many streams, we focused on the stream which included both STON2 and the previously reported keratoconus susceptibility genes. Since we focused on this stream, we could narrow down the candidate genes from 42 genes to 7 genes. By evaluating the association of these seven genes, we could identify the additional novel keratoconus susceptibility gene SMAD3. If we had not focused on this stream, we might not have identified SMAD3. As discussed above, it was necessary to combine GWAS and WDD in this study.
Changes in the Manuscript
-

Reviewer's comment
p151312: I'm convinced by the GWAS strategy, but not convinced the mouse model, and since the relationship between the GWAS outcomes <-> non-KC mouse model <-> computational analysis has several assumptions, and a small N of KC cases, I'm puzzled. The presence of STON2 in a mouse model is not directly linked to a human-KC model, or human-KC samples (acquired after corneal transplant surgery for instance).
Response to Reviewer
Thank you for your comment. We understand the reviewer's concern. In the immunostaining portion, we only confirmed the expression of STON2 in mouse corneal tissue and agree that the evaluation of gene expression in human KC tissue will be required to reveal a more detailed role of STON2 in keratoconus development. However, since the expression of STON2 has never been evaluated in human KC tissue, disease KC mouse model nor healthy mouse cornea, we believe the confirmation of STON2 expression in healthy mouse cornea can serve as a first step towards further immunohistological evaluation of STON2. Additionally, from a statistical point of view, due to a large number of control samples, we could estimate population allele frequency of healthy individuals precisely enough to detect statistically-significant differences, despite the fact that estimation of population allele frequency of keratoconus individuals was not precise due to small sample size. Although we understand that the current result is not confirmatory and should be further replicated in the future, we believe these results suggest the contribution of STON2 in the development of keratoconus. We hope that the contribution of STON2 on keratoconus will be further investigated in the future, inspired by the current results.
Changes in the Manuscript
-

Reviewer's comment
The WDD analysis to me appears as an independent method to identify pathways, and it's relationship with the authors' GWAS is unclear to me.
Response to Reviewer
Thank you for your comment. As we responded above, by using WDD we tried to determine the association between STON2 with previously reported keratoconus susceptibility genes by analyzing the FOXO1-SMAD3-STON2 stream. If we had not focused on this stream, we might have not paid attention to SMAD3. We, therefore, believe that the WDD section is not independent of the GWAS.
Changes in the Manuscript
-

Reviewer's comment
p151320: this is indeed a major strength.
Response to Reviewer
Thank you for your comment.
Changes in the Manuscript
-

Reviewer's comment
p16l335: these downstream effects could be attributed to other cases as well, most notably physical damage by eye-rubbing, or immunological changes (please refer to https://www.ncbi.nlm.nih.gov/pubmed/26235733 in this aspect). The current line of reasoning is to direct for me.
Response to Reviewer
Thank you for the important comment. As you noted, changes such as thinning of the corneal stroma, breakages in Bowman's layer, and degeneration of the corneal epithelium can be attributed to physical damage by eye-rubbing, or immunological changes. STON2 might be associated with vulnerability to eye-rubbing or immunological changes. We cited the article and revised the manuscript accordingly.
Changes in the Manuscript
The eQTL data revealed that the effect size of rs2371597 on STON2 expression was strongest in skeletal muscle, which is rich in collagen, a protein that reportedly plays a key role in keratoconus pathogenesis. Although no previous reports have investigated the expression of STON2 in human corneal tissue, we speculate that STON2 may play an important role in keratoconus development by interacting with extracellular matrix remodelling. Our immunohistochemical staining results for STON2 are compatible with a previous histopathological study of keratoconus, which showed that thinning of the corneal stroma, breaks in Bowman's layer, and degeneration of the corneal epithelium were the characteristics of corneas in patients with keratoconus. ^{2,23} STON2 might be associated with the vulnerability to physical damage or immunological changes. Although biological studies examining the role of STON2 in human corneal tissue are required to prove our hypothesis, pathways involving STON2 may serve as novel targets for treating keratoconus by controlling basal cell degeneration. (page 10-11, line 204-217)

Reviewer's comment
p18l385: here the conclusion is framed much more nuanced and to -the -point.
Response to Reviewer
Thank you for your comment. We revised the section.
Changes in the Manuscript
In summary, we identified two novel keratoconus susceptibility loci STON2 and SMAD3 by integrating conventional GWAS and artificial intelligence, using WDD. Cognitive-computing technology combined with GWAS can assist in identifying hidden relationships among disease-susceptibility genes and potential susceptibility genes, enabling more efficient interpretation of GWAS results. We believe that the current approach can be generalized for application to numerous other diseases. Since samples from patients with a disease are more difficult to obtain than samples from healthy individuals, which can be collected through cohort studies, the current approach will prove particularly helpful in facilitating the exploration of disease-susceptible genes. (page 13, line 264-272)

Reviewer #3

Reviewer's comment
This article reports the results of a GWAS for central corneal thickness (CCT) in a sample of 3584 healthy Japanese volunteers (Nagahama cohort). Of the two genome-wide significant loci identified, one was already known to be associated with CCT (FNDC3B) while the other was novel (STON2; lead SNP rs2371597). Association of STON2 SNP rs2371597 with CCT was replicated in cohorts of Malay, Chinese, and Indian ethnicity recruited from Singapore. This variant was also found to be associated with keratoconus in a case-control sample from Japan (179 cases, 11084 controls; OR=1.27, P=0.04) and shown to be an eQTL for STON2 in some GTEx tissues. STON2 was expressed in basal corneal epithelial cells of mouse cornea. Furthermore, a bioinformatics literature mining analysis using the IBM Watson Drug Discovery algorithm identified SMAD3 as an additional candidate gene for keratoconus. In support of this theory, SNP rs12912547 in SMAD3 was associated with keratoconus in the case-control sample (R=1.44, P=0.001). In general these findings are interesting and have both fundamental mechanistic relevance and clinical relevance. The GWAS component of the manuscript is scientifically convincing, whereas more clarification is required to demonstrate the validity of the Watson Drug Discovery analysis. These points are covered in more detail below.
Response to Reviewer
Thank you for your comment.
Changes in the Manuscript
-

Reviewer's comment
Specific points 1. L52. The recent GWAS for keratoconus by Hardcastle et al. ARVO abstract # 4249 (2019) should be cited.
Response to Reviewer
Thank you for your comment. We cited the abstract as you recommended.
Changes in the Manuscript
Although previous genome-wide association studies (GWASs) were performed with keratoconus patients, no genetic region with a significant genome-wide association has been identified thus far.⁵⁻⁷ (page 4, line 50-52)

Reviewer's comment
2. L133. Were any of the 2942 participants in the Nagahama replication sample related to any of the 3584 participants in the Nagahama discovery GWAS sample? If so, then this is not independent replication; the related individuals should be excluded.
Response to Reviewer
Thank you for your comment. Since the genome-wide genotype data of 2,942 samples are not available, we cannot eliminate the possibility that some of these individuals are somehow related to the participants of the discovery GWAS. However, since the association was replicated in other cohorts, we believe that the association results would not be greatly biased.
Changes in the Manuscript
-

Reviewer's comment

3. L154. It is unusual to include only 1 PC as a GWAS covariate; typically 5, 10 or 20 PCs are included. Please justify this a priori choice of 1 PC and report if the 2 genome-wide significant associations from the discovery GWAS were altered if adjustment was made for 10 or 20 PCs.

Response to Reviewer

Thank you for your comment. As an inflation factor (λ_{GC}) of 1.065 indicated acceptable control of the study population substructure, we chose only 1 PC for the GWAS.

We performed GWAS again adjusting for 10 and 20 PCs as you recommended and found only two genetic loci showed genome wide significant association with CCT, the same as analysis adjusted for 1 PC. The Manhattan plots are shown below (Figure 1 and Figure 2).

In summary, *STON2* rs2371597 showed a genome wide significance level of association with corneal thickness adjusted for 10 PCs ($\beta = 5.44$, $P = 9.76 \times 10^{-12}$) and 20 PCs ($\beta = 5.42$, $P = 9.76 \times 10^{-11}$). *FNDC3B* rs4894538 also showed genome wide significance level association with corneal thickness adjusted for 10 PCs ($\beta = -4.14$, $P = 1.95 \times 10^{-8}$) and 20 PCs ($\beta = -4.08$, $P = 3.04 \times 10^{-8}$). We, therefore, confirmed the result of our GWAS adjusted for only 1 PC was robust.

Figure 1. Manhattan plot of QTL data from GWAS of CCT adjusted for age, sex, axial length and 10 PCs

Reviewer's comment
4. L158. Give details of the fixed effects meta-analysis weighting, e.g. inverse variance.
Response to Reviewer
We apologize for not providing a sufficient description in the original manuscript. We used inverse variance weighted fixed effect model and clarified this in the revised manuscript.
Changes in the Manuscript
Experimental-wide significance in the discovery stage was set at $P < 5.0 \times 10^{-8}$. All meta-analyses were performed using an inverse variance weighted fixed effect model . Thereafter, differences were considered statistically significant at $P < 0.05$. (page 16, line 348-350)

Reviewer's comment
5. L171. State explicitly if the cases and controls were genotyped using the same genotyping platform and if the imputation pipelines used for cases and controls were exactly the same. If cases and controls were genotyped separately, this can lead to spurious association signals. How did the authors assess whether associations were truly due to disease status and not simply a technical artefact?
Response to Reviewer
Thank you for the comment. First, I will clarify the genotyping and imputation here. (1) Both cases and controls from Yokohama City University were genotyped using the same genotyping platform, OmniExpress, and imputed using the same pipelines. (2) A part of control samples from Nagahama Cohort were genotyped using a series of BeadChip DNA arrays, namely the HumanHap610 Quad, HumanOmni2.5-4, HumanOmni2.5-8, HumanOmni2.5s, CoreExome24, and HumanExome arrays as shown in Method section. The rest of the samples from Nagahama Cohort were genotyped using TaqMan SNP genotyping assay. (3) Control samples from Tohoku Medical Megabank Project were derived from a publicly available database. Since MAF of rs2371597 are consistent across these three control groups (Yokohama City University: 0.248, Nagahama cohort: 0.244, Tohoku Medical Megabank Project: 0.256, respectively) and consistently low compared with keratoconus patients (0.293) showing the statistically significant difference, we believe it was not a technical artifact but a true signal. We clarified this in the revised manuscript.
Changes in the Manuscript
A detailed description of these cohorts and the genotyping methods used are provided in the Supplementary Note. Briefly, both cases and controls were recruited from the Yokohama City University, and genotyped using the same genotyping platform, OmniExpress, and imputed using the same pipelines. Control samples from the Nagahama Cohort were genotyped as described above. Control samples from the Tohoku Medical Megabank Project were derived from a publicly available database. Differences in the genotype distributions of these cohorts were tested using the Cochran–Armitage test. $P < 0.05$ was set as the threshold of statistical significance. (page 17, line 363-370)

Reviewer's comment
6. L171. Which covariates were included in the case-control GWAS? Were any participants related?
Response to Reviewer
Thank you for the comment. Regarding the case-control analysis of keratoconus, we did not perform GWAS. Instead, candidate gene analysis was performed to evaluate the association of CCT-associated loci with keratoconus. Chi-squared test for trend without adjustment was applied for this analysis. Here, because we could not access individual-level genome-wide genotype data except for a part of Nagahama Study, we could not estimate the pairwise relatedness of whole 11,263 individuals. Thus, despite estimated close relatives were excluded in each dataset where the genome-wide data were available (shown in supplementary note), we could not eliminate the cryptic relatedness within the whole 11,263 individuals. Still, because MAF of rs2371597 are consistent across these three control groups (Yokohama City University: 0.248, Nagahama cohort: 0.244, Tohoku Medical Megabank Project: 0.256, respectively) and these three site are geographically apart (Yokohama City University in Tokyo area, Nagahama city in Kyoto area, and Tohoku Medical Megabank in Tohoku area), we believe the effect of the residual relatedness was minimal.
Changes in the Manuscript
-

Reviewer's comment
7. L204. Cite references for the studies describing these known risk SNPs.
Response to Reviewer
Thank you for the comment. We cited the studies according to the reviewer's recommendation.
Changes in the Manuscript
To detect additional keratoconus-susceptibility loci and infer pathways associated with keratoconus, we performed WDD predictive analysis. Eight previously reported keratoconus-susceptibility genes (FNDC3B, COL5A1, FOXO1, MPDZ, NF1B, RXRA, BANP, and ZNF469) and the newly identified keratoconus susceptibility gene were input as 'teaching genes'. ^{12,20,21} (page 18, line 394-398)

Reviewer's comment
8. L217. More detail is required on the assessment of candidate genes identified by the WDD analysis. Please report (a) how many genes in total were identified in the WDD analysis, (b) were all of these genes assessed in the keratoconus case-control sample? (If only genes in the FOXO1-SMAD stream were assessed this may appear to be cherry-picking) (c) was 1 SNP examined per gene? (d) was the reported p-value of p=0.001 corrected for multiple testing?
Response to Reviewer
We apologize for not describing this process clearly in the original manuscript as I believe that this has led to some confusion. To be precise, WDD was not used to identify candidate genes, but to narrow down candidate genes. Analysis flow was as follows; firstly, we selected 53 genes which showed mild association with central corneal thickness ($P < 0.0001$) as candidate genes for susceptibility to keratoconus. Secondly, we assessed the relationship between the 53 genes and known keratoconus susceptibility genes using WDD, which generated Figure 4. Lastly, we focused on only the stream including STON2 since its identification as a keratoconus susceptible gene was the novel finding of the current study. Considering the above, we would like to answer your question as follows: (a) As explained, WDD was not used to identify candidate genes, but to narrow down candidate genes. (b) As mentioned above, we assessed only seven genes (NRXN1, CPLX2, CSMD1, ADAM12, SMAD3, WWOX, CDH13) within a stream including STON2. We do not believe that this is cherry-picking as there is a clear criterion applied. To say more, we applied WDD so as to not need to evaluate all genes. (c) We examined only one top SNP with the lowest P value in discovery CCT GWAS per gene. (d) The reported p-value of $p = 0.001$ is nominal. Significance level was set as 0.0071 (0.05/7) considering multiple testing correction, since we evaluated only seven SNPs here. Thus, we reported it was a significant association after correcting for multiple testing. If we had taken a more traditional approach (i.e. a strategy without using WDD), a more stringent cut-off of 0.00094 (0.05/53) would have been applied. Thanks to the novel strategy using WDD, we were able to reduce the curse of multiplicity.
Changes in the Manuscript
-

Reviewer's comment
9. L281. Does an eQTL database exist for cornea or corneal epithelium? If so, please report results for the lead STON2 SNP or a surrogate. If not, please mention this lack of a suitably matched eQTL database in the text.
Response to Reviewer
Thank you for your comment. Unfortunately, an eQTL database for cornea or corneal epithelium does not exist. We mentioned the lack of eQTL data in human corneal tissue in the revised manuscript.
Changes in the Manuscript
A search of a publicly available expression quantitative trait loci analysis (eQTL) database revealed that rs2371597 was significantly associated with STON2 expression (GTEx Portal; https://gtexportal.org/home/). A multi-tissue eQTL plot revealed that the normalised effect size (NES) of rs2371597 on STON2 expression was strongest in the skeletal muscle (NES = -0.189, P = 1.3×10^{-8}; Figure 2, https://www.gtexportal.org/home/snp/rs2371597) in which collagen plays an important role in providing its tensile strength and elasticity. However, data on the association of rs2371597 with gene expression in human corneal tissue was not available. Our immunohistochemical study of mouse corneas showed that STON2 was expressed in the corneal epithelial cell layer (Figure 2). Our results demonstrated that strong STON2 expression mainly occurred in basal cells rather than superficial cells in the corneal epithelium. In the stroma and endothelium layer, only minimal STON2 expression was observed. (page 7-8, line 137-149)

Reviewer's comment
10. L349. Test for shared molecular mechanisms between keratoconus and schizophrenia using LD score regression.
Response to Reviewer
Thank you for your recommendation. We agree that LD score regression would provide useful information about shared molecular mechanism. However, since it may be outside the scope of the current study, we plan to perform such analysis in our future study. Thank you so much for your advice.
Changes in the Manuscript
-

Reviewer's comment																																																																				
11. Table 1. Instead of the "Nearby gene" column, it would be helpful to state whether the SNP is genic, intronic, etc.																																																																				
Response to Reviewer																																																																				
Thank you for the suggestion. We included these information in Table 1.																																																																				
Changes in the Manuscript																																																																				
   SNP Chr Position Effect allele EAF Gene Discovery stage Replication stage Meta-analysis   N β P N β P N β P     rs4894538 3 171999005 T 0.299 FNDC3B (intron) 3584 -4.13 1.90×10^{-8} - - - - - -   rs2371597 14 81873377 C 0.242 STON2 (intron) 3584 5.35 1.91×10^{-11} 2942 2.95 3.58×10^{-4} 6526 4.20 2.32×10^{-13}   															SNP	Chr	Position	Effect allele	EAF	Gene	Discovery stage			Replication stage			Meta-analysis			N	β	P	N	β	P	N	β	P	rs4894538	3	171999005	T	0.299	FNDC3B (intron)	3584	-4.13	1.90×10^{-8}	-	-	-	-	-	-	rs2371597	14	81873377	C	0.242	STON2 (intron)	3584	5.35	1.91×10^{-11}	2942	2.95	3.58×10^{-4}	6526	4.20	2.32×10^{-13}
SNP	Chr	Position	Effect allele	EAF	Gene	Discovery stage			Replication stage			Meta-analysis																																																								
						N	β	P	N	β	P	N	β	P																																																						
rs4894538	3	171999005	T	0.299	FNDC3B (intron)	3584	-4.13	1.90×10^{-8}	-	-	-	-	-	-																																																						
rs2371597	14	81873377	C	0.242	STON2 (intron)	3584	5.35	1.91×10^{-11}	2942	2.95	3.58×10^{-4}	6526	4.20	2.32×10^{-13}																																																						
Chr, chromosome; EAF, effect allele frequency; β , beta; P , P-value																																																																				

Reviewer's comment
12. Table 3 could be omitted and the salient information presented in the text.
Response to Reviewer
Thank you for your advice. We omitted the Table 3 and revised the manuscript.
Changes in the Manuscript
To evaluate the possible association between STON2 rs2371597 and keratoconus, we conducted a case-controlled study using 179 keratoconus patients, regardless of age, sex or physical impairment, from the Yokohama City University and 11,084 Japanese healthy controls. This analysis revealed that minor-allele frequency (MAF) at rs2371597 C was significantly high in keratoconus patients than that in controls (MAF_{keratoconus} = 0.293, MAF_{controls} = 0.246, odds ratio [OR] (95% confidence interval [CI]) = 1.27 (1.01 - 1.60), P = 0.041). (Page 7, line 128-134)

Reviewer's comment
13. Figure1 and 2. The small font size made the text difficult to read.
Response to Reviewer
Thank you for bringing this to our attention. We revised Figure 1 to improve the readability. As it is difficult to revise Figure 2, we also provided the link for Figure 2 instead.
Changes in the Manuscript
A multi-tissue eQTL plot revealed that the normalised effect size (NES) of rs2371597 on STON2 expression was strongest in the skeletal muscle (NES = -0.189, P = 1.3×10^{-8} ; Figure 2, https://www.gtexportal.org/home/snp/rs2371597) in which collagen plays an important role in providing its tensile strength and elasticity. However, data on the association of rs2371597 with gene expression in human corneal tissue was not available. (page 8, line 139-144)

Reviewer's comment
14. Figure 3 legend. Define blue label for nuclei. 15. Figure 5. Define symbols. 16. General. Please present 95% CI when reporting odds ratios.
Response to Reviewer
Thank you so much for the careful review. We revised the manuscript according to the reviewer's suggestion throughout the manuscript.
Changes in the Manuscript
Figure 3. Expression of STON2 in retinal ganglion cells of C57BL/6 mice. Mouse corneal sections were immunostained with an antibody against STON2 (green). The nuclei are counterstained with 40,6-diamidino-2-phenylindole (blue) in the merged image. The upper portion of the panel shows the corneal epithelium. Figure 5. Scatter plot of WDD-based predictive analysis. A scatter plot of genes was generated based on their distances in the literature. Nine teacher genes (green triangles) and 42 candidate genes (red circles) are shown. (Figure legends)

Reviewer's comment
Grammatical and typographical suggestions 1. Abstract L33. Use "heritable" in place of "inheritable".
Response to Reviewer
Thank you for this comment. We revised the sentence.
Changes in the Manuscript
Central corneal thickness (CCT) is a highly in heritable characteristic that is associated with keratoconus. (abstract, line 33)

Reviewer's comment
2. Abstract L34. I suggest, "...we identified a novel locus for CCT, STON2 rs2371597..."
Response to Reviewer
Thank you for your comment. As stated in the title, the novel point of the current study was to identify novel keratoconus susceptibility genes. Since the association of STON2 and CCT was recently reported in another paper, we did not use the word "novel" in this sentence.
Changes in the Manuscript
-

Reviewer's comment
3. Abstract L35. Correctly format of p-value.
Response to Reviewer
We revised the sentence.
Changes in the Manuscript
namely STON2 rs2371597 ($P = 2.32 \times 10^{-13}$). (abstract, line 35)

Reviewer's comment
3. Abstract L35. Correctly format of p-value. 4. L142. The meaning of "one open data" is not clear. Please re-word. 5. L157. I suggest, "...experiment-wide significance..." 6. L286. What is meant by "remarkable expression". Please re-word. 7. L302. Define "ToMMo".
Response to Reviewer
Thank you so much for the careful review. We revised the sentence according to the reviewer's suggestion.
Changes in the Manuscript
3 namely STON2 rs2371597 ($P = 2.32 \times 10^{-13}$). (abstract, line 35) 4 We also included publicly available data on a Latino population (n = 3,584). Detailed information and the sample-collection methods for these cohorts can be found in the Supplementary Table 6 and Supplementary Note and in a previous publication.¹⁷ (page 16, line 336-337) 5 Experimenta-wide significance in the discovery stage was set at $P < 5.0 \times 10^{-8}$. (page 16, line 348) 6 Our immunohistochemical study of mouse corneas showed that STON2 was expressed in the corneal epithelial cell layer (Figure 2). Our results demonstrated that strong STON2 expression mainly occurred in basal cells rather than superficial cells in the corneal epithelium. In the stroma and endothelium layer, only minimal STON2 expression was observed. (page 8, line 145-149) 7 Since the SMAD3 rs11333560 genotype was not available in the Tohoku Medical Megabank Project, we instead employed a proxy SNP (rs12913547 in SMAD3). (page 9, line 163-164)

REVIEWERS' COMMENTS:

Reviewer #1 (Remarks to the Author):

Thank you very much to the authors for their comprehensive responses to my previous comments. Reading the responses to other reviewers' comments, I noticed a certain inconsistency. In the manuscript, the authors wrote that keratoconus is a common disease. However, in response to comments regarding the number of studied patients, the authors indicated that this is a rare disease. I understand that this is not a disease that is as common as hypertension or type 2 diabetes, but I suggest using more uniform terminology in the future.

Reviewer #2 (Remarks to the Author):

The authors have answered all stated questions in detail and made appropriate alterations when deemed necessary.

Their approach is an example to others and I have nothing further to add.

I now better understand the interrelatedness of the GWAS and WDD and think this is a very valuable paper on keratoconus genetics.

Reviewer #3 (Remarks to the Author):

The authors have addressed or rebutted my previous comments. Two additional points merit attention:

Line 88. Explain in the text that inclusion of additional PCs minimally altered the findings.

Line 175. The following sentence is problematic: "Immunohistochemical staining of mouse cornea revealed that STON2 expression was localised to corneal basal cells, indicating that STON2 served an important role in keratoconus development." The site of expression does not provide evidence of a causal role. In contrast, the GWAS does support this inference.

Reviewers' comments:

Reviewer #1

Reviewer's comment
Thank you very much to the authors for their comprehensive responses to my previous comments. Reading the responses to other reviewers' comments, I noticed a certain inconsistency. In the manuscript, the authors wrote that keratoconus is a common disease. However, in response to comments regarding the number of studied patients, the authors indicated that this is a rare disease. I understand that this is not a disease that is as common as hypertension or type 2 diabetes, but I suggest using more uniform terminology in the future.
Response to Reviewer
Thank you for your comments.
Changes in the Manuscript
-

Reviewer #2

Reviewer's comment
The authors have answered all stated questions in detail and made appropriate alterations when deemed necessary. Their approach is an example to others and I have nothing further to add. I now better understand the interrelatedness of the GWAS and WDD and think this is a very valuable paper on keratoconus genetics.
Response to Reviewer
Thank you for your comment. We believe our approach may provide new approach for further genetic studies.
Changes in the Manuscript
-

Reviewer #3

Reviewer's comment

The authors have addressed or rebutted my previous comments. Two additional points merit attention:

Line 88. Explain in the text that inclusion of additional PCs minimally altered the findings.

Line 175. The following sentence is problematic: "Immunohistochemical staining of mouse cornea revealed that STON2 expression was localised to corneal basal cells, indicating that STON2 served an important role in keratoconus development." The site of expression does not provide evidence of a causal role. In contrast, the GWAS does support this inference.

Response to Reviewer

Thank you for your comment. We revised the manuscript as you recommended.

Changes in the Manuscript

An inflation factor (λ_{GC}) of 1.065 indicated good control of the study population substructure (Supplementary Figure 1). **We also found that inclusion of additional PCs minimally altered the findings.**

(page 4, line 88-89)

We found that the STON2 rs2371597 allele was significantly associated with CCT and keratoconus, as well as significantly altered STON2 expression in human tissues.

~~Immunohistochemical staining of mouse cornea revealed that STON2 expression was localised to corneal basal cells, indicating that STON2 served an important role in keratoconus development.~~ WDD analysis assisted in identifying another **novel** keratoconus-susceptibility SNP, i.e. SMAD3 rs12913547.

(page 4, line 173-177)